# DYffusion: A Dynamics-informed Diffusion Model for Spatiotemporal Forecasting

**Salva Rühling Cachay**      **Bo Zhao**      **Hailey Joren**      **Rose Yu**

University of California, San Diego

{sruhlingcachay, bozhao, hjoren, roseyu}@ucsd.edu

## Abstract

While diffusion models can successfully generate data and make predictions, they are predominantly designed for static images. We propose an approach for efficiently training diffusion models for probabilistic spatiotemporal forecasting, where generating stable and accurate rollout forecasts remains challenging, Our method, DYffusion, leverages the temporal dynamics in the data, directly coupling it with the diffusion steps in the model. We train a stochastic, time-conditioned interpolator and a forecaster network that mimic the forward and reverse processes of standard diffusion models, respectively. DYffusion naturally facilitates multi-step and long-range forecasting, allowing for highly flexible, continuous-time sampling trajectories and the ability to trade-off performance with accelerated sampling at inference time. In addition, the dynamics-informed diffusion process in DYffusion imposes a strong inductive bias and significantly improves computational efficiency compared to traditional Gaussian noise-based diffusion models. Our approach performs competitively on probabilistic forecasting of complex dynamics in sea surface temperatures, Navier-Stokes flows, and spring mesh systems.[1]

## 1  Introduction

Dynamics forecasting refers to the task of predicting the future behavior of a dynamic system, involving learning the underlying dynamics governing the system's evolution to make accurate predictions about its future states. Obtaining accurate and reliable probabilistic forecasts is an important component of policy formulation [39, 4], risk management [38, 20], resource optimization [7], and strategic planning [46, 72]. In many applications, accurate long-range probabilistic forecasts are particularly challenging to obtain [38, 20, 4]. In operational settings, methods typically hinge on sophisticated numerical simulations that require supercomputers to perform calculations within manageable time frames, often compromising the spatial resolution of the grid for efficiency [3].

Generative modeling presents a promising avenue for probabilistic dynamics forecasting. Diffusion models, in particular, can successfully model natural image and video distributions [47, 54, 59]. The standard approach, Gaussian diffusion, corrupts the data with Gaussian noise to varying degrees through the *"forward process"* and sequentially denoises a random input at inference time to obtain highly realistic samples through the *"reverse process"* [60, 25, 32]. However, learning to map from noise to realistic data is challenging in high dimensions, especially with limited data availability. Consequently, the computational costs associated with training and inference from diffusion models are prohibitive, requiring a sequential sampling process over hundreds of diffusion steps. For example, a denoising diffusion probabilistic model (DDPM) takes around 20 hours to sample 50k images of size $32 \times 32$ [61]. In addition, few methods apply diffusion models beyond static images. Video

---

[1]Code is available at: https://github.com/Rose-STL-Lab/dyffusion

37th Conference on Neural Information Processing Systems (NeurIPS 2023).

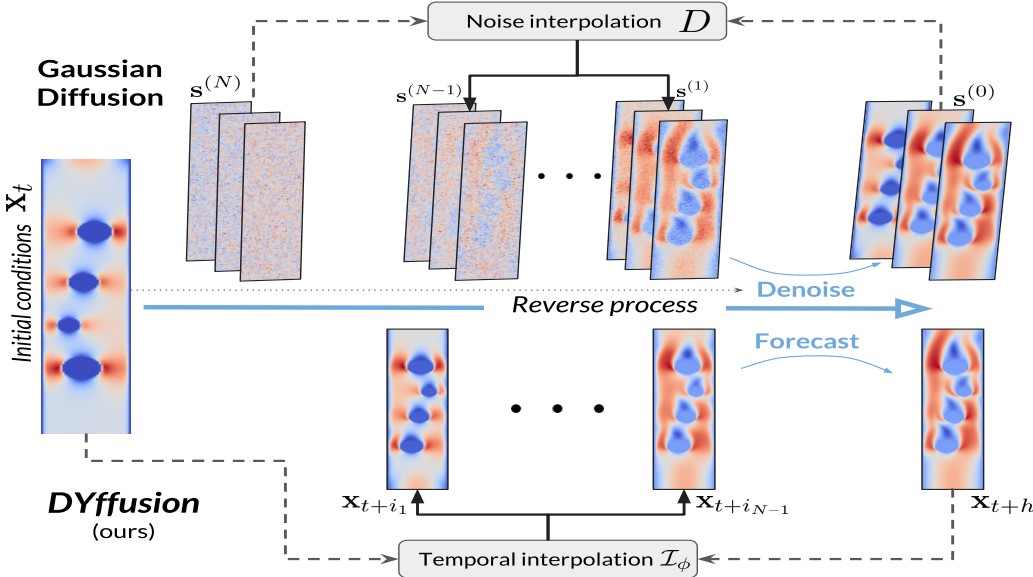

Figure 1: Our proposed framework, DYffusion, reimagines the noise-denoise forward-backward processes of conventional diffusion models as an interplay of temporal interpolation and forecasting. On the top row, we illustrate the direct application of a video diffusion model to dynamics forecasting for a horizon of $h = 3$. On the bottom row, DYffusion generates continuous-time probabilistic forecasts for $\mathbf{x}_{t+1:t+h}$, given the initial conditions, $\mathbf{x}_t$. During sampling, the reverse process iteratively steps forward in time by forecasting $\mathbf{x}_{t+h}$ (which plays the role of the "clean data", $\mathbf{s}^{(0)}$, in conventional diffusion models) and interpolating to one of $N$ intermediate timesteps, $\mathbf{x}_{t+i_n}$. As a result, our approach operates in the data space at all times and does not need to model high-dimensional videos at each diffusion state.

diffusion models [27, 24, 77, 67] can generate realistic samples but do not explicitly leverage the temporal nature of the data to generate accurate forecasts.

In this work, we introduce a novel framework to train a dynamics-informed diffusion model for multi-step probabilistic forecasting. Inspired by recent findings that highlight the potential of non-Gaussian diffusion processes [2, 28, 10], we introduce a new forward process. This process relies on temporal interpolation and is implemented through a time-conditioned neural network. Without requiring assumptions about the physical system, our approach imposes an inductive bias by coupling the diffusion process steps with the time steps in the dynamical system. This reduces the computational complexity of our diffusion model in terms of memory footprint, data efficiency, and the number of diffusion steps required during training. Our resulting diffusion model-based framework, which we call DYffusion, naturally captures long-range dependencies and generates accurate probabilistic ensemble forecasts for high-dimensional spatiotemporal data.

Our contributions can be summarized as follows:

- We investigate probabilistic spatiotemporal forecasting from the perspective of diffusion models, including their applications to complex physical systems with a large number of dimensions and low data availability.

- We introduce DYffusion, a flexible framework for multi-step forecasting and long-range horizons that leverages a temporal inductive bias to accelerate training and lower memory needs. We explore the theoretical implications of our method, proving that DYffusion is an implicit model that learns the solutions to a dynamical system, and cold sampling [2] can be interpreted as its Euler's method solution.

- We conduct an empirical study comparing performance and computational requirements in dynamics forecasting, including state-of-the-art probabilistic methods such as conditional video diffusion models. We find that the proposed approach achieves strong probabilistic forecasts and improves computational efficiency over standard Gaussian diffusion.

## 2 Background

**Problem setup.** We study the problem of probabilistic spatiotemporal forecasting. We start with a dataset of $\{\mathbf{x}_t\}_{t=1}^T$ snapshots with $\mathbf{x}_t \in \mathcal{X}$. Here, $\mathcal{X}$ represents the space in which the data lies, which may consist of spatial dimensions (e.g., latitude, longitude, atmospheric height) and a channel dimension (e.g., velocities, temperature, humidity). The task of probabilistic forecasting is to learn a conditional distribution $P(\mathbf{x}_{t+1:t+h} \mid \mathbf{x}_{t-l+1:t})$ that uses $l$ snapshots from the past to forecast a subsequent *horizon* of $h$ snapshots. Here, we focus on the task of forecasting from a single initial condition, where we aim to learn $P(\mathbf{x}_{t+1:t+h} \mid \mathbf{x}_t)$. This covers many realistic settings [50, 44] while minimizing the computational requirements for accurate forecasts [65].

**Diffusion processes.** In a standard diffusion model, the *forward diffusion process* iteratively degrades the data with increasing levels of (Gaussian) noise. The reverse diffusion process then gradually removes the noise to generate data. We denote these diffusion step states, $\mathbf{s}^{(n)}$, with a superscript $n$ to clearly distinguish them from the time steps of the data $\{\mathbf{x}_t\}_{t=1}^T$. This process can be generalized to consider a degradation operator $D$ that takes as input the data point $\mathbf{s}^{(0)}$ and outputs $\mathbf{s}^{(n)} = D(\mathbf{s}^{(0)}, n)$ for varying degrees of degradation proportional to $n \in \{1, \ldots, N\}$ [2]. Oftentimes, $D$ adds Gaussian noise with increasing levels of variance so that $\mathbf{s}^{(N)} \sim \mathcal{N}(\mathbf{0}, \mathbf{I})$. A denoising network parameterized by $\theta$, $R_\theta$ is trained to *restore* $\mathbf{s}^{(0)}$, i.e. such that $R_\theta(\mathbf{s}^{(n)}, n) \approx \mathbf{s}^{(0)}$. The diffusion model can be conditioned on the input dynamics by considering $R_\theta(\mathbf{s}^{(n)}, \mathbf{x}_t, n)$. In the case of dynamics forecasting, the diffusion model can be trained to minimize the objective

$$\min_\theta \mathbb{E}_{n \sim \mathcal{U}[\![1,N]\!], \mathbf{x}_t, \mathbf{s}^{(0)} \sim \mathcal{X}} \left[ ||R_\theta(D(\mathbf{s}^{(0)}, n), \mathbf{x}_t, n) - \mathbf{s}^{(0)}||^2 \right], \tag{1}$$

where $\mathcal{X}$ is the data distribution, $|| \cdot ||$ is a norm, and $\mathbf{s}^{(0)} = \mathbf{x}_{t+1:t+h}$ is the prediction target. $\mathcal{U}[\![a, b]\!]$ describes the uniform distribution over the set $\{a, a+1, \cdots, b\}$. In practice, $R_\theta$ may be trained to predict the Gaussian noise that has been added to the data point using score matching objective [25]. The objective in Eq. (1) can be viewed as a generalized version of the standard diffusion models, see Appendix B.2 for a more comprehensive analysis.

## 3 DYffusion: DYnamics-Informed Diffusion Model

The key innovation of our method, DYffusion, is a reimagining of the diffusion processes to more naturally model spatiotemporal sequences, $\mathbf{x}_{t:t+h}$ (see Fig. 1). Specifically, we replace the degradation operator, $D$, with a stochastic interpolator network $\mathcal{I}_\phi$, and the restoration network, $R_\theta$, with a deterministic forecaster network, $F_\theta$.

---

**Algorithm 1** DYffusion, Two-stage Training

**Input:** networks $F_\theta, \mathcal{I}_\phi$, norm $|| \cdot ||$, horizon $h$, schedule $[i_n]_{i=0}^{N-1}$
*Stage 1:* Train interpolator network, $\mathcal{I}_\phi$
    1. Sample $i \sim \mathtt{Uniform}(\{1, \ldots, h-1\})$
    2. Sample $\mathbf{x}_t, \mathbf{x}_{t+i}, \mathbf{x}_{t+h} \sim \mathcal{X}$
    3. Optimize $\min_\phi ||\mathcal{I}_\phi(\mathbf{x}_t, \mathbf{x}_{t+h}, i) - \mathbf{x}_{t+i}||^2$

*Stage 2:* Train forecaster network (diffusion model backbone), $F_\theta$
    1. Freeze $\mathcal{I}_\phi$ and enable inference stochasticity (e.g. dropout)
    2. Sample $n \sim \mathtt{Uniform}(\{0, \ldots, N-1\})$ and $\mathbf{x}_t, \mathbf{x}_{t+h} \sim \mathcal{X}$
    3. Optimize $\min_\theta ||F_\theta(\mathcal{I}_\phi(\mathbf{x}_t, \mathbf{x}_{t+h}, i_n), i_n) - \mathbf{x}_{t+h}||^2$

---

At a high level, our forward and reverse processes emulate the temporal dynamics in the data. Thus, intermediate steps in the diffusion process can be reused as forecasts for actual timesteps in multi-step forecasting. Another benefit of our approach is that the reverse process is initialized with the initial conditions of the dynamics and operates in observation space at all times. In contrast, a standard diffusion model is designed for unconditional generation, and reversing from white noise requires more diffusion steps. For conditional prediction tasks such as forecasting, DYffusion emerges as a much more natural method that is well aligned with the task at hand. See Table 4 for a full glossary.

**Temporal interpolation as a forward process.** To impose a temporal bias, we train a time-conditioned network $\mathcal{I}_\phi$ to interpolate between snapshots of data. Specifically, given a horizon $h$, we train $\mathcal{I}_\phi$ so that $\mathcal{I}_\phi(\mathbf{x}_t, \mathbf{x}_{t+h}, i) \approx \mathbf{x}_{t+i}$ for $i \in \{1, \ldots, h-1\}$ using the objective:

$$\min_\phi \mathbb{E}_{i \sim \mathcal{U}[\![1,h-1]\!], \mathbf{x}_{t,t+i,t+h} \sim \mathcal{X}} \left[ ||\mathcal{I}_\phi(\mathbf{x}_t, \mathbf{x}_{t+h}, i) - \mathbf{x}_{t+i}||^2 \right]. \tag{2}$$

Interpolation is an easier task than forecasting, and we can use the resulting interpolator $\mathcal{I}_\phi$ during inference to interpolate beyond the temporal resolution of the data. That is, the time input can be continuous, with $i \in (0, h-1)$, where we note that the range $(0, 1)$ is outside the training regime, as discussed in Appendix D.5.6. To generate probabilistic forecasts, it is crucial for the interpolator, $\mathcal{I}_\phi$, to *produce stochastic outputs* within the diffusion model and during inference time. We enable this using Monte Carlo dropout [17] at inference time.

**Forecasting as a reverse process.** In the second stage, we train a forecaster network $F_\theta$ to forecast $\mathbf{x}_{t+h}$ such that $F_\theta(\mathcal{I}_\phi(\mathbf{x}_t, \mathbf{x}_{t+h}, i|\xi), i) \approx \mathbf{x}_{t+h}$ for $i \in S = [i_n]_{n=0}^{N-1}$, where $S$ denotes a schedule mapping the diffusion step to the interpolation timestep. The interpolator network, $\mathcal{I}_\phi$, is frozen with inference stochasticity enabled, represented by the random variable $\xi$. Here, $\xi$ stands for the randomly dropped out weights of the neural network and is omitted henceforth for clarity. Specifically, we seek to optimize the objective

$$\min_\theta \mathbb{E}_{n \sim \mathcal{U}[\![0, N-1]\!], \mathbf{x}_{t, t+h} \sim \mathcal{X}} \left[ ||F_\theta(\mathcal{I}_\phi(\mathbf{x}_t, \mathbf{x}_{t+h}, i_n|\xi), i_n) - \mathbf{x}_{t+h}||^2 \right]. \tag{3}$$

To include the setting where $F_\theta$ learns to forecast the initial conditions, we define $i_0 := 0$ and $\mathcal{I}_\phi(\mathbf{x}_t, \cdot, i_0) := \mathbf{x}_t$. In the simplest case, the forecaster network is supervised by all possible timesteps given by the temporal resolution of the training data. That is, $N = h$ and $S = [j]_{j=0}^{h-1}$. Generally, the interpolation timesteps should satisfy $0 = i_0 < i_n < i_m < h$ for $0 < n < m \le N-1$. Given the equivalent roles between our forecaster and the denoising net in diffusion models, we also refer to them as the diffusion backbones. As the time condition to our diffusion backbone is $i_n$ instead of $n$, we can choose *any* diffusion-dynamics schedule during training or inference and even use $F_\theta$ for unseen timesteps (see Fig. 3). The resulting two-stage training algorithm is shown in Alg. 1.

Because the interpolator $\mathcal{I}_\phi$ is frozen in the second stage, the imperfect forecasts $\hat{\mathbf{x}}_{t+h} = F_\theta(\mathcal{I}_\phi(\mathbf{x}_t, \mathbf{x}_{t+h}, i_n), i_n)$ may degrade accuracy when used during sequential sampling. To handle this, we introduce a one-step look-ahead loss term $||F_\theta(\mathcal{I}_\phi(\mathbf{x}_t, \hat{\mathbf{x}}_{t+h}, i_{n+1}), i_{n+1}) - \mathbf{x}_{t+h}||^2$ whenever $n + 1 < N$ and weight the two loss terms equally. Additionally, providing a clean or noised form of the initial conditions $\mathbf{x}_t$ as an additional input to the forecaster net can improve performance. These additional tricks are discussed in Appendix B.

**Sampling.** Taken together, we can write the generative process of DYffusion as follows:

$$p_\theta(\mathbf{s}^{(n+1)}|\mathbf{s}^{(n)}, \mathbf{x}_t) = \begin{cases} F_\theta(\mathbf{s}^{(n)}, i_n) & \text{if } n = N-1 \\ \mathcal{I}_\phi(\mathbf{x}_t, F_\theta(\mathbf{s}^{(n)}, i_n), i_{n+1}) & \text{otherwise,} \end{cases} \tag{4}$$

where $\mathbf{s}^{(0)} = \mathbf{x}_t$ and $\mathbf{s}^{(n)} \approx \mathbf{x}_{t+i_n}$ correspond to the initial conditions and predictions of intermediate steps, respectively. In our formulations, we reverse the diffusion step indexing to align with the temporal indexing of the data. That is, $n = 0$ refers to the start of the reverse process (i.e. $\mathbf{x}_t$), while $n = N$ refers to the final output of the reverse process (here, $\mathbf{x}_{t+h}$). Our reverse process steps forward in time, in contrast to the mapping from noise to data in standard diffusion models. As a result, DYffusion should require fewer diffusion steps and data.

Similarly to the forward process in [61, 2], our interpolation stage ceases to be a diffusion process. Our forecasting stage as detailed in Eq. (3), follows the (generalized) diffusion model objectives. This similarity allows us to use many existing diffusion model sampling methods for inference, such as the cold sampling algorithm

---

**Algorithm 2** Adapted Cold Sampling [2] for DYffusion

1: **Input:** Initial conditions $\hat{\mathbf{x}}_t := \mathbf{x}_t$, schedule $[i_n]_{i=0}^{N-1}$, output timesteps $J$ (by default $J = \{1, \ldots, h-1\}$)
2: **for** $n = 0, 1, \ldots, N-1$ **do**
3: $\quad \hat{\mathbf{x}}_{t+h} \leftarrow F_\theta(\hat{\mathbf{x}}_{t+i_n}, i_n)$
4: $\quad \hat{\mathbf{x}}_{t+i_{n+1}} = \mathcal{I}_\phi(\mathbf{x}_t, \hat{\mathbf{x}}_{t+h}, i_{n+1}) - \mathcal{I}_\phi(\mathbf{x}_t, \hat{\mathbf{x}}_{t+h}, i_n) + \hat{\mathbf{x}}_{t+i_n}$
5: **end for**
6: $\hat{\mathbf{x}}_{t+j} \leftarrow \mathcal{I}_\phi(\mathbf{x}_t, \hat{\mathbf{x}}_{t+h}, j), \forall j \in J$    # Optional refinement
7: **Return:** $\{\hat{\mathbf{x}}_{t+j} \mid j \in J\} \cup \{\hat{\mathbf{x}}_{t+h}\}$

---

from [2] (see Alg. 2). In Appendix C.2, we also discuss a simpler but less performant sampling algorithm. During the sampling process, our method essentially alternates between forecasting and interpolation, as illustrated in Fig. 2. $F_\theta$ always predicts the last timestep, $\mathbf{x}_{t+h}$, but iteratively improves those forecasts as the reverse process comes closer in time to $t + h$. This is analogous to the iterative denoising of the "clean" data in standard diffusion models. This motivates line 6

of Alg. 2, where the final forecast of $\mathbf{x}_{t+h}$ can be used to fine-tune intermediate predictions or to increase the temporal resolution of the forecast. DYffusion can be applied autoregressively to forecast even longer rollouts beyond the training horizon, as demonstrated by the Navier-Stokes and spring mesh experiments in section 5.

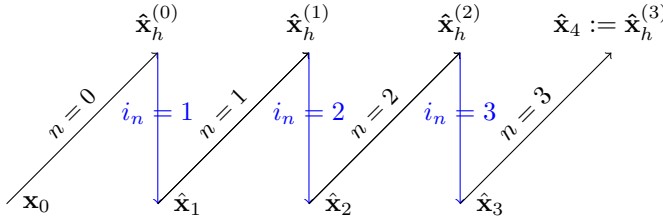

Figure 2: During sampling, DYffusion alternates between forecasting and interpolation, following Alg. 2. In this example, the sampling trajectory follows a simple schedule of going through all integer timesteps that precede the horizon of $h = 4$, with the number of diffusion steps $N = h$. The output of the last diffusion step is used as the final forecast for $\hat{\mathbf{x}}_4$. The **black** lines represent forecasts by the forecaster network, $F_\theta$. The first forecast is based on the initial conditions, $\mathbf{x}_0$. The **blue** lines represent the subsequent temporal interpolations performed by the interpolator network, $\mathcal{I}_\phi$.

**Memory footprint.**   DYffusion requires only $\mathbf{x}_t$ and $\mathbf{x}_{t+h}$ (plus $\mathbf{x}_{t+i}$ during the first stage) to train, resulting in a constant memory footprint as a function of $h$. In contrast, direct multi-step prediction models including video diffusion models or (autoregressive) multi-step loss approaches require $\mathbf{x}_{t:t+h}$ to compute the loss. This means that these models must fit $h + 1$ timesteps of data into memory, which scales poorly with the forecasting horizon $h$. Therefore, many are limited to predicting a small number of frames. MCVD, for example, trains on a maximum of 5 video frames due to GPU memory constraints [67].

**Reverse process as ODE.**   DYffusion can be interpreted as modeling the dynamics by a "diffusion process", similar to an implicit model [13]. This view helps explain DYffusion's superior forecasting ability, as it learns to model the physical dynamical system $\mathbf{x}(t)$. The neural networks $\mathcal{I}_\phi$ and $F_\theta$ are trained to construct a differential equation that implicitly models the solution to $\mathbf{x}(t)$.

Let $s$ be the time variable. DYffusion models the dynamics as

$$\frac{d\mathbf{x}(s)}{ds} = \frac{d\mathcal{I}_\phi\left(\mathbf{x}_t, F_\theta(\mathbf{x}, s), s\right)}{ds}. \tag{5}$$

The initial condition is given by $\mathbf{x}(t) = \mathbf{x}_t$. During prediction, we seek to obtain

$$\mathbf{x}(s) = \mathbf{x}(t) + \int_t^s \frac{d\mathcal{I}_\phi\left(\mathbf{x}_t, F_\theta(\mathbf{x}, s), s\right)}{ds} ds \qquad \text{for } s \in (t, t + h]. \tag{6}$$

The prediction process is equivalent to evaluating (6) at discrete points in the forecasting window $(t, t + h]$. Sampling schedules $i_n$ can be thought of as step sizes in ODE solvers. Therefore, different sampling schedules can be flexibly chosen at inference time, since they are discretizations of the same ODE. Different sampling algorithms can be obtained from different ODE solvers and we prove that cold sampling is an approximation of the Euler method (derivation in Appendix C.1).

Viewing DYffusion as an implicit model emphasizes its dynamics-informed nature and reveals the relation between DYffusion and DDIM [61]. Specifically, DDIM can be viewed as an implicit model that models the solution to the noise-removing dynamics from a random image to a clean image. In comparison, DYffusion is an implicit model that learns the solutions of a dynamical system using data from the current step $\mathbf{x}_t$ to the future step $\mathbf{x}_{t+h}$.

## 4   Related Work

**Diffusion Models.**   Diffusion models [25, 60, 62, 63] have demonstrated significant success in diverse domains, notably images and text [76]. While interesting lines of work explore Gaussian diffusion for multivariate time-series imputation [1] and forecasting [51], high-dimensional spatiotemporal

forecasting has remained unexplored. Methods such as combining blurring and noising techniques [28, 10, 2] and adopting a unified perspective with Poisson Flow Generative Models [75] have shown promise as alternative approaches to extend diffusion processes beyond Gaussian noise. Although intellectually intriguing, the results of cold diffusion [2] and other recent attempts that use different corruptions [53] are usually inferior to Gaussian diffusion. Our work departs from these methods in that we propose a dynamics-informed process for the conditional generation of high-dimensional forecasts that is not based on multiple levels of data corruption.

**Diffusion Models for Video Prediction.** Most closely related to our work are diffusion models for videos [67, 27, 77, 59, 26, 24]. The task of conditional video prediction can be placed under the multi-step notation of Eq. (1). Notably, we are interested in modeling the full dynamics and the underlying physics rather than object-centric tasks. There is also evidence that techniques tailored to computer vision and NLP do not transfer well to spatiotemporal problems [66]. These video diffusion models also rely internally on the standard Gaussian noise diffusion process inherited from their image-generation counterparts. As such, these models miss the opportunity to incorporate the temporal dynamics of videos into the diffusion process.

**Dynamics Forecasting.** Deterministic next-step forecasting models can be unrolled autoregressively to achieve multi-step predictions. However, this may result in poor or unstable rollouts due to compounding prediction errors [11, 57, 9, 8, 34, 5, 33, 43]. The issue can be alleviated by unrolling the model at training time to introduce a loss term on the multi-step predictions [45, 34, 36, 23, 5]. However, this approach is memory-intensive and comes at the expense of efficiency since the model needs to be called sequentially for each multi-step loss term. It can also introduce new instabilities because the gradients have to be computed recursively through the unrolled steps. Even then, this approach may fall behind physics-based baselines for long-range horizons [45]. Alternatively, forecasting multiple steps all at once as in video diffusion models [73, 52, 5] or conditioned on the time [37, 49, 15, 43], training separate models for each required horizon [56, 22, 68], noising the training data [55], and encoding physics knowledge into the ML model [11, 14, 70, 41, 9, 35, 8], can often produce better long-range forecasts, though they still require a fixed temporal resolution. For probabilistic dynamics forecasting [74], most methods focus on autoregressive next-step prediction models [58, 18, 29]. Some notable works target precipitation nowcasting [52], or short-term forecasting [15]. Alternatively to using intrinsically probabilistic models, one can use a deterministic model to generate an ensemble forecast by perturbing its initial conditions [45, 21]. Our framework addresses primary challenges in the field [64], specifically realistic long-range forecasts and efficient modeling of multi-step dynamics.

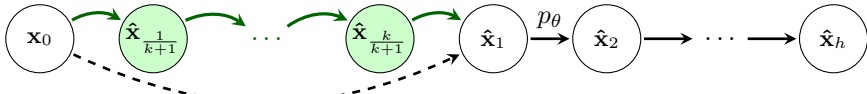

Figure 3: Example schedules for coupling diffusion steps to dynamical time steps. While a naive approach only uses timesteps given by the temporal resolution of the data (i.e. discrete indices), our framework can accommodate continuous indices. The additional $k$ diffusion steps are highlighted in **green** and map uniformly between the input timestep, $\mathbf{x}_0$, and earliest output timestep, $\hat{\mathbf{x}}_1$. Our experiments using the SST dataset in section 5 demonstrate that increasing the number of diffusion steps with implicit intermediate timesteps can improve performance (see Appendix D.7).

## 5 Experiments

### 5.1 Datasets

We evaluate our method and baselines on three different datasets:

- **Sea Surface Temperatures (SST):** a new dataset based on NOAA OISSTv2 [30], which comes at a daily time-scale. Similar to [11, 71], we train our models on regional patches which increases the available data (here, we choose 11 boxes of 60 latitude ×60 longitude resolution in the eastern tropical Pacific Ocean). Unlike the data based on the NEMO dataset in [11, 71], we choose OISSTv2

Table 1: Results for sea surface temperature forecasting of 1 to 7 days ahead, as well Navier-Stokes flow full trajectory forecasting of 64 timesteps. Numbers are averaged out over the evaluation horizon. **Bold** indicates best, blue second best. For CRPS and MSE, lower is better. For SSR, closer to 1 is better. For SST, all models are trained on forecasting $h = 7$ timesteps. The time column represents the time needed to forecast all 7 timesteps for a single batch. For Navier-Stokes, Perturbation, Dropout, and DYffusion are trained on a horizon of $h = 16$. MCVD and DDPM are trained on $h = 4$ and $h = 1$, respectively, as we were not able to successfully train them using larger horizons.

| Method | SST | | | | Navier-Stokes | | |
|---|---|---|---|---|---|---|---|
| | CRPS | MSE | SSR | Time [s] | CRPS | MSE | SSR |
| Perturbation | $0.281 \pm 0.004$ | $0.180 \pm 0.011$ | $0.411 \pm 0.046$ | 0.4241 | $0.090 \pm 0.001$ | $0.028 \pm 0.000$ | $0.448 \pm 0.002$ |
| Dropout | $0.267 \pm 0.003$ | $0.164 \pm 0.004$ | $0.406 \pm 0.042$ | 0.4241 | $0.078 \pm 0.001$ | $0.027 \pm 0.001$ | $0.715 \pm 0.005$ |
| DDPM | $0.246 \pm 0.005$ | $0.177 \pm 0.005$ | $0.674 \pm 0.011$ | 0.3054 | $0.180 \pm 0.004$ | $0.105 \pm 0.010$ | $0.573 \pm 0.001$ |
| MCVD | **0.216** | **0.161** | 0.926 | 79.167 | $0.154 \pm 0.043$ | $0.070 \pm 0.033$ | $0.524 \pm 0.064$ |
| DYffusion | $0.224 \pm 0.001$ | $0.173 \pm 0.001$ | **1.033** $\pm$ **0.005** | 4.6722 | **0.067** $\pm$ **0.003** | **0.022** $\pm$ **0.002** | **0.877** $\pm$ **0.006** |

as our SST dataset because it contains more data (although it has a lower spatial resolution of $1/4°$ compared to $1/12°$ of NEMO). We train, validate, and test all models for the years 1982-2019, 2020, and 2021, respectively. The preprocessing details are in the Appendix D.1.1.

- **Navier-Stokes flow:** benchmark dataset from [44], which consists of a $221 \times 42$ grid. Each trajectory contains four randomly generated circular obstacles that block the flow. The viscosity is $1e$-3. The channels consist of the $x$ and $y$ velocities as well as a pressure field. Boundary conditions and obstacle masks are given as additional inputs to all models.

- **Spring Mesh:** benchmark dataset from [44]. It represents a $10 \times 10$ grid of particles connected by springs, each with mass 1. The channels consist of two position and momentum fields each.

## 5.2 Experimental Setup

We focus on *probabilistic multi-step forecasting*, and especially long rollouts, as opposed to attaining the best next-step or short-term forecasts. This is an important distinction because, while deterministic single-step forecasting models such as those of [44] are appropriate for short-term predictive skill comparison, such models tend to lack stability for long-term forecasts. We demonstrate this in the Appendix D.4.2 (see Fig. 6), where our method and baselines excel in the long-range forecasting regime while being competitive in the short-range regime compared to the baselines from [44].

**Neural network architectures.** For a given dataset, we use the *same backbone architecture* for all baselines as well as for both the interpolation and forecaster networks in DYffusion. For the SST dataset, we use a popular UNet architecture designed for diffusion models[2]. For the Navier-Stokes and spring mesh datasets, we use the UNet and CNN from the original paper [44], respectively. The UNet and CNN models from [44] are extended by the sine/cosine-based featurization module of the SST UNet to embed the diffusion step or dynamical timestep.

**Baselines.** We compare our method against both direct applications of standard diffusion models to dynamics forecasting and methods to ensemble the "barebone" backbone network of each dataset. The network operating in "barebone" form means that there is no involvement of diffusion.

- **DDPM [25]:** both next-step (image-like problem) as well as multi-step (video-like problem) prediction mode (we report the best model version only in the results).

- **MCVD [67]:** conditional video diffusion model for video prediction (in "concat" mode).

- **Dropout [17]:** Ensemble multi-step forecasting based on enabling dropout of the barebone backbone network at inference time.

- **Perturbation [45]:** Ensemble multi-step forecasting based on random perturbations of the initial conditions/inputs with a fixed variance of the barebone backbone network.

---

[2]https://github.com/lucidrains/denoising-diffusion-pytorch/blob/main/denoising_diffusion_pytorch/denoising_diffusion_pytorch.py

Table 2: Spring Mesh results. Both methods are trained on a horizon of $h = 134$ timesteps and evaluated how well they forecast the full test trajectories of 804 steps. Despite several attempts with varying configurations over the number of diffusion steps, learning rates, and diffusion schedules, none of the DDPM or MCVD diffusion models converged. Using the same CNN architecture, our MSE results significantly surpass the ones reported in Fig. 8 of [44], where the CNN diverged or attained a very poor MSE. This is likely because of the multi-step training approach that we use, while the single-step prediction approach ($h = 1$) in [44] can generate unstable rollouts.

| Method | Test | | | Out of Distribution | | |
|--------|------|------|------|---------------------|------|------|
| | CRPS | MSE | SSR | CRPS | MSE | SSR |
| Dropout | $0.0138 \pm 0.0006$ | $7.27\text{e-}4 \pm 6.8\text{e-}5$ | $\mathbf{1.01} \pm \mathbf{0.02}$ | $0.0448 \pm 0.0007$ | $7.08\text{e-}3 \pm 1.7\text{e-}4$ | $0.70 \pm 0.01$ |
| DYffusion | $\mathbf{0.0103} \pm \mathbf{0.0022}$ | $\mathbf{4.20\text{e-}4} \pm \mathbf{2.1\text{e-}4}$ | $1.13 \pm 0.08$ | $\mathbf{0.0292} \pm \mathbf{0.0009}$ | $\mathbf{3.54\text{e-}3} \pm \mathbf{2.0\text{e-}4}$ | $\mathbf{0.82} \pm \mathbf{0.00}$ |

MCVD and the multi-step DDPM variant predict the timesteps $\mathbf{x}_{t+1:t+h}$ based on $\mathbf{x}_t$, which is the standard approach in video (prediction) diffusion models. The barebone backbone network baselines are time-conditioned forecasters (similarly to the DYffusion forecaster) trained on the multi-step objective $\mathbb{E}_{i \sim \mathcal{U}[\![1,h]\!], \mathbf{x}_{t,t+i} \sim \mathcal{X}} ||F_\theta(\mathbf{x}_t, i) - \mathbf{x}_{t+i}||^2$ from scratch. We found it to perform very similarly to predicting all $h$ horizon timesteps at once in a single forward pass (not shown). In all experiments, we observed that the single-step forecasting ($h = 1$) version of the barebone network yielded significantly lower performance compared to any of the multi-step training approaches (see Appendix D.4.2). More details of the implementation are provided in Appendix D.2.

**Evaluation.** We evaluate the models on the best validation Continuous Ranked Probability Score (CRPS) [42], which is a proper scoring rule and a popular metric in the probabilistic forecasting literature [19, 12, 51, 48, 58]. The CRPS is computed by generating a 20-member ensemble (i.e. 20 samples are drawn per batch element), while we generate a 50-member ensemble for final model selection between different hyperparameter runs. We also use a 50-member ensemble for evaluation on the test datasets to compute the CRPS, mean squared error (MSE), and spread-skill ratio (SSR). The MSE is computed on the ensemble mean prediction. The SSR is defined as the ratio of the square root of the ensemble variance to the corresponding ensemble RMSE. It serves as a measure of the reliability of the ensemble, where values smaller than 1 indicate underdispersion (i.e. the probabilistic forecast is overconfident in its forecasts), and larger values overdispersion [16, 18]. On the Navier-Stokes and spring mesh datasets, models are evaluated by autogressively forecasting the full test trajectories of length 64 and 804, respectively. For the SST dataset, all models are evaluated on forecasts of up to 7 days. We do not explore more long-term SST forecasts because the chaotic nature of the system, and the fact that we only use regional patches, inherently limits predictability.

## 5.3 Results

**Quantitative Results.** We present the time-averaged metrics for the SST and Navier-Stokes dataset in Table 1 and for the Spring Mesh dataset in Table 2. DYffusion performs best on the Navier-Stokes dataset, while coming in a close second on the SST dataset after MCVD, in terms of CRPS. Since MCVD uses 1000 diffusion steps[3], it is slower to sample from at inference time than from our DYffusion which is trained with less than 50 diffusion steps. The DDPM model for the SST dataset is fairly efficient because it only uses 5 diffusion steps but lags in terms of performance. Thanks to the time-conditioned nature of DYffusion's interpolator and forecaster nets, memory is not an issue when scaling our framework to long horizons. In the spring mesh dataset, we train with a horizon of 134 and evaluate long trajectories of 804 steps. Our method beats the barebone network baseline, with a larger margin on the out-of-distribution test dataset. It is worth noting that our reported MSE scores are significantly better than the ones reported in [44], likely due to multi-step training being superior to the single-step forecasting approach of [44] (see Appendix D.4.2 for more details). For the out-of-distribution test set of the Navier-Stokes benchmark, the results are almost identical to the one in Tab. 1, so we show them in Appendix D.4.1.

**Qualitative Results.** In dynamics forecasting, long-range forecasts of ML models often suffer from blurriness (or might even diverge when using autoregressive models). In Fig. 4 we show exemplary

---

[3]This is the default, we were not able to successfully train MCVD models with fewer diffusion steps.

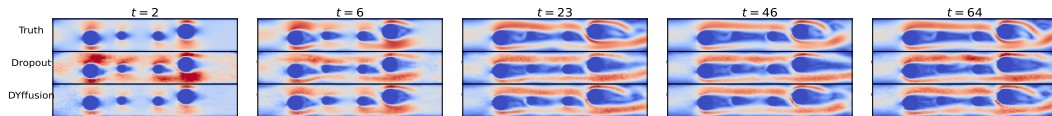

Figure 4: Qualitative forecasts for timesteps 2, 6, 24, 46, and 64 (last timestep) of the velocity norm of an example Navier-Stokes test trajectory. Here, we generate five sample trajectories for the best baseline (Dropout) and our method DYffusion, both with $h = 16$, and visualize the one with the best trajectory-average MSE for each of the methods. Our method (bottom row) can reproduce fine-scale details visibly better than the baseline (see e.g. right sides of the snapshots). The corresponding video of the full trajectory, including the velocity and pressure fields, can be found at this Google Drive URL `https://drive.google.com/file/d/1xklVs42Ii18I8SVTOf1ZmAKR159qiHG_/view?usp=share_link`.

samples for the best baseline (Dropout) and DYffusion as well as the corresponding ground truth at five different timesteps from a complete Navier-Stokes trajectory forecast. Our method can reproduce the true dynamics over the full trajectory and does so better than the baseline, especially for fine-scale patterns such as the tails of the flow after the right-most obstacle. The corresponding full video can be found at this URL.

**Increasing the forecasted resolution.** Motivated by the continuous-time nature of DYffusion, we aim to study in this experiment whether it is possible to forecast skillfully beyond the resolution given by the data. Here, we forecast the same Navier-Stokes trajectory shown in Fig. 4 but at $8\times$ resolution (i.e. 512 timesteps instead of 64 are forecasted in total). This behavior can be achieved by either changing the sampling trajectory $[i_n]_{n=0}^{N-1}$ or by including additional output timesteps, $J$, for the refinement step of Alg. 2. In this case, we choose to do the latter and find the resulting forecast to be visibly pleasing and temporally consistent; see the full video at this URL.

Note that we hope that our probabilistic forecasting model can capture any of the possible, uncertain futures instead of forecasting their mean, as a deterministic model would do. As a result, some long-term rollout samples are expected to deviate from the ground truth. For example, see the velocity at $t = 3.70$ in the video. It is reassuring that our samples show sufficient variation, but also cover the ground truth quite well (sample 1).

**Ablations.** In Appendix D.5, we ablate various components of our method. This includes showing that inference stochasticity in the interpolation network is crucial for performance. Our experiments in Appendix D.5.1 confirm the findings of [2] regarding the superiority of cold sampling (see Alg. 2) compared to the naive sampling (Alg. 4) algorithm. We provide a theoretical justification in Appendix C.2, where we show that cold sampling has first-order discretization error, whereas naive sampling does not.

Table 3: CRPS test scores for the SST dataset using different number of training years. 2, 4, 37 refers to training from 2017, 2015, 1982 until the end of 2018, respectively. The standard deviations are around 0.01 or lower.

| # years | 2 | 4 | 37 |
|---|---|---|---|
| MCVD | 0.262 | 0.233 | 0.216 |
| DYffusion | 0.239 | 0.234 | 0.225 |

We also study the choice of the interpolation schedule:
For the Navier-Stokes and spring mesh datasets, it was sufficient to use the simplest possible schedule $S = [i_n]_{n=0}^{N-1} = [j]_{j=0}^{h-1}$, where the number of diffusion steps and the horizon are equal. For the SST dataset, using $k$ additional diffusion steps corresponding to floating-point $i_k$, especially such that $i_k < 1$ for all $k$, improved performance significantly. Similarly to the finding of DDIM [61] for Gaussian diffusion, we show that the sampling trajectories of DYffusion can be accelerated by skipping intermediate diffusion steps, resulting in a clear trade-off between accuracy and speed. Finally, we verify that the forecaster's prediction of $\mathbf{x}_{t+h}$ for a given diffusion step $n$ and associated input $\hat{\mathbf{x}}_{t+i_n}$ usually improves as $n$ increases, i.e. as we approach the end of the reverse process and $\mathbf{x}_{t+i_n}$ gets closer in time to $\mathbf{x}_{t+h}$.

The variance in the performance of the standard diffusion models across datasets may be intriguing. One possible explanation could be the size of the training dataset, since the SST dataset is more than ten times larger than the Navier-Stokes and spring mesh datasets, respectively. Sufficient training data may be a key factor that enables Gaussian diffusion models to effectively learn the reverse process

from noise to the data space. We explore this hypothesis in Table 3, where we retrain MCVD as well as DYffusion's $F_\theta$ and $\mathcal{I}_\phi$ on small subsets of the SST training set. We find that DYffusion's performance degrades more gracefully under limited training data, especially when using only two years of data (or $\approx 8,000$ training data points).

We found it very challenging to train video diffusion models on long horizons. A reason could be that the channel dimension of the video diffusion backbone is scaled by the training horizon, which increases the problem dimensionality and complexity considerably (see Appendix D.6). Unlike common practice in natural image or video problems, we do not threshold the predictions of any diffusion model, as the data range is unbounded. However, this thresholding has been shown to be an important implementation detail that stabilizes diffusion model sampling for complex problems in the domain of natural images [40]. This could be the reason why the studied diffusion model baselines produce poor predictions for the Navier-Stokes and spring mesh datasets.

## 6 Conclusion

We have introduced DYffusion, a novel dynamics-informed diffusion model for improved probabilistic spatiotemporal forecasting. In contrast to most prior work on probabilistic forecasting that often generates long-range forecasts via iterating through an autoregressive model, we tailor diffusion models to naturally support long-range forecasts during inference time by coupling their diffusion process with the dynamical nature of the data. Our study presents the first comprehensive evaluation of diffusion models for the task of spatiotemporal forecasting. Finally, we believe that our dynamics-informed diffusion model can serve as a more general source of inspiration for tailoring diffusion models to other conditional generation problems, such as super-resolution, where the data itself can natively inform an inductive bias for the diffusion model.

**Limitations.** While our proposed framework can be applied to any dynamics forecasting problem as an alternative to autoregressive models, it does not support, in its current form, problems where the output space is different from the input space. Additionally, although it requires fewer diffusion steps compared to Gaussian diffusion models to achieve similar performance, the need for additional forward passes through the interpolator network diminishes some of these advantages. When prioritizing inference speed, approaches that only require a single forward pass to forecast outperform sequential sampling of diffusion models, including DYffusion.

**Future work.** Cold Diffusion [2] indicates that deterministic data degradations cannot meet the performance of stochastic (i.e. Gaussian noise-based) degradations. In our work, we have experimented with using an interpolator network with Monte Carlo dropout enabled during inference time to achieve a stochastic forward process, which we have shown to be important for optimal performance. Exploring more advanced approaches for introducing stochasticity into our general framework presents an interesting avenue for future work. A more advanced approach could draw insights from the hypernetwork-based parameterization of the latent space from [69]. Furthermore, given our finding of the equivalence of cold sampling with the Euler solver (C.1), it could be interesting to consider more advanced ODE solvers for improved sampling from DYffusion. Recent advances in neural architectures for physical systems [6, 31] are complementary to our work. Using such alternative neural architectures as interpolator or forecaster networks within our framework could be a promising approach to further improve performance.

## Acknowledgements

This work was supported in part by the U.S. Army Research Office under Army-ECASE award W911NF-07-R-0003-03, the U.S. Department Of Energy, Office of Science, IARPA HAYSTAC Program, NSF Grants #2205093, #2146343, and #2134274.

NOAA OI SST V2 High Resolution Dataset data provided by the NOAA PSL, Boulder, Colorado, USA, from their website at `https://psl.noaa.gov`.

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

# Contents

# Appendix

## A Glossary

The glossary is given in Table 4 below.

| Symbol | Used for |
|---|---|
| $t$ | Indexer of temporal data snapshots |
| $h$ | Training horizon, i.e. the number of timesteps a method learns to forecast |
| $H$ | Evaluation horizon. A model with horizon $h$, is applied autoregressively $\lceil \frac{H}{h} \rceil$-times during evaluation |
| $N$ | Total number of diffusion steps |
| $n$ | Indexer of the diffusion step, where $0 \leq n \leq N - 1$ |
| $\mathcal{X}$ | Distribution from which the data snapshots are sampled from |
| $\mathbf{x}_{t+i}$ | Data point at timestep $t + i$ |
| $\hat{\mathbf{x}}_{t+i}$ | Predicted data point at timestep $t + i$ |
| $\mathbf{x}_{t:t+h}$ | Shorthand for the sequence of data points $\mathbf{x}_t, \mathbf{x}_{t+1}, \ldots, \mathbf{x}_{t+h}$ |
| $\mathbf{x}_t$ | Initial conditions, i.e. input snapshot for the forecasting model |
| $\mathbf{x}_{t+1:t+h}$ | Sequence of target snapshots that the forecasting model predicts given $\mathbf{x}_t$ |
| $\mathbf{s}^{(n)}$ | Data point at diffusion step $n$ |
| $\mathcal{I}_\phi$ | Stochastic temporal interpolation neural network; Trained in the first-stage of DYffusion |
| $F_\theta$ | Forecasting neural network; Trained in the second-stage of DYffusion |
| $S$ | Interpolation schedule for DYffusion that couples every diffusion step, $n$, to a timestep $i_n$ |
| $i_n$ | Maps diffusion step $n$ to timestep $i_n$, where $0 = i_0 < i_n < i_m < h$ for $0 < n < m \leq N - 1$ |
| $k$ | Number of non-integer auxiliary diffusion steps for DYffusion. If $k = 0$, then $S = [i_n]_{n=0}^{N-1} = [j]_{j=0}^{h-1}$ |
| $D$ | Forward diffusion process operator, e.g. Gaussian noise, of a conventional diffusion model |
| $R_\theta$ | Denoising neural network of a conventional diffusion model |
| $\mathcal{U}[\![a, b]\!]$ | Describes the uniform distribution over the set $\{a, a+1, \cdots, b\}$ |

Table 4: Glossary of variables and symbols used in this paper. Note that for DYffusion, the reverse process starts at $n = 0$ and progresses forward (together with the corresponding interpolation timestep, $i_n$) as $n \to N$ ending with the final output $\mathbf{s}^{(N)}$. For conventional diffusion models, this notation is reversed and the final output of the reverse diffusion process corresponds to $\mathbf{s}^{(0)}$.

## B Methodology

### B.1 DYffusion

---
**Algorithm 3** DYffusion, Two-stage Training, all details
---

**Input:** networks $F_\theta, \mathcal{I}_\phi$, norm $||\cdot||$, horizon $h$, schedule $[i_n]_{i=0}^{N-1}$, loss coefficients $\lambda_1, \lambda_2$, forecaster conditioning $\mathbf{c}$

*Stage 1:* Train interpolator network, $\mathcal{I}_\phi$
    1. Sample $i \sim \texttt{Uniform}\left(\{1, \ldots, h - 1\}\right)$
    2. Sample $\mathbf{x}_t, \mathbf{x}_{t+i}, \mathbf{x}_{t+h} \sim \mathcal{X}$
    3. Optimize $\min_\phi ||\mathcal{I}_\phi\left(\mathbf{x}_t, \mathbf{x}_{t+h}, i\right) - \mathbf{x}_{t+i}||^2$

*Stage 2:* Train forecaster network (diffusion model backbone), $F_\theta$
    1. Freeze $\mathcal{I}_\phi$ and enable inference stochasticity (e.g. dropout)
    2. Sample $n \sim \texttt{Uniform}\left(\{0, \ldots, N - 1\}\right)$ and $\mathbf{x}_t, \mathbf{x}_{t+h} \sim \mathcal{X}$
    3. $\hat{\mathbf{x}}_{t+h}^{(1)} \leftarrow F_\theta(\mathcal{I}_\phi\left(\mathbf{x}_t, \mathbf{x}_{t+h}, i_n\right), i_n, \mathbf{c}(\mathbf{x}_t, n))$
    4. $\hat{\mathbf{x}}_{t+h}^{(2)} \leftarrow F_\theta(\mathcal{I}_\phi\left(\mathbf{x}_t, \hat{\mathbf{x}}_{t+h}^{(1)}, i_{n+1}\right), i_{n+1}, \mathbf{c}(\mathbf{x}_t, n+1))$ `if` $n \leq N - 2$ `else` $\mathbf{x}_{t+h}$
    5. Optimize $\min_\theta \lambda_1 ||\hat{\mathbf{x}}_{t+h}^{(1)} - \mathbf{x}_{t+h}||^2 + \lambda_2 ||\hat{\mathbf{x}}_{t+h}^{(2)} - \mathbf{x}_{t+h}||^2$

---

In Alg. 3 we give the full training algorithm for DYffusion. The only differences to the truncated training algorithm in the main text (Alg. 1) are that we also include:

- One-step look-ahead loss term trick, $\lambda_2||F_\theta(\mathcal{I}_\phi(\mathbf{x}_t, \hat{\mathbf{x}}_{t+h}, i_{n+1}), i_{n+1}) - \mathbf{x}_{t+h}||^2$, whenever $n+1 < N$. Essentially, the look-ahead loss term simulates one step of the sampling process (Alg. 2) and backpropagates through it, so that the network is trained with an objective that is closer to and partially mimics the iterative sampling process. In all experiments, we use $\lambda_1 = \lambda_2 = \frac{1}{2}$. We believe that it would be interesting to explore alternative approaches to this loss term, such as fine-tuning the interpolator network in the second-stage too.

- Optional conditioning of the forecaster network on a noised or clean version of the initial conditions, $\mathbf{x}_t$. We explore three different approaches: 1) No conditioning at all, i.e. $\mathbf{c}(\cdot, \cdot) = \texttt{null}$; 2) Clean initial conditions, i.e. $\mathbf{c}(\mathbf{x}_t, \cdot) = \mathbf{x}_t$; 3) Noised initial conditions, where we sample random noise $\epsilon \sim \mathcal{N}(\mathbf{0}, \mathbf{I})$ of the same shape of $\mathbf{x}_t$, and then define $\mathbf{c}(\mathbf{x}_t, n) = \frac{n}{N-1}\mathbf{x}_t + (1 - \frac{n}{N-1})\epsilon$. The second variant is analogous to the conditioning of conventional diffusion models, which DYffusion does not necessarily require since the initial conditions indirectly influence the main input of the forecaster net (i.e. $\hat{\mathbf{x}}_{t+i_n} = \mathcal{I}_\phi(\mathbf{x}_t, \mathbf{x}_{t+h}, i_n)$). The motivation for the third variant stems from trying to give the forecaster net a more explicit conditioning on the initial conditions when the reverse process goes farther away from $t$ (i.e. as $n \to N$ and $\hat{\mathbf{x}}_{t+i_n} \to \mathbf{x}_{t+h}$). An ablation for the choice of $\mathbf{c}$ is reported in D.5.4.

### B.2 Connection to standard diffusion models

Our work builds upon cold diffusion [2], which "paves the way for generalized diffusion models that invert arbitrary processes". The cold sampling algorithm, which we use to sample from DYffusion (see Alg. 2), is a generalization of DDIM sampling for "generalized diffusion models" (see Appendix A.6 of [2]) and a key ingredient to make DYffusion work (see D.5.1). Our proposed forward and reverse processes are specifically designed to fall under this framework. Given this context, it becomes clear that DYffusion is a generative model for forecasting that falls in the category of "generalized diffusion models".

**Training objectives.** To see the similarity between standard diffusion models and DYffusion in terms of training objectives, we write out the corresponding equations Eq. 1 and Eq. 3. For a standard "generalized" diffusion model, we can rewrite the main objective inside Eq. 1 as follows:

$$||R_\theta(D(\mathbf{s}^{(0)}, n), \mathbf{x}_t, n) - \mathbf{s}^{(0)}||^2 = ||R_\theta(\mathbf{s}^{(n)}, \mathbf{x}_t, n) - \mathbf{s}^{(0)}||^2, \tag{7}$$

where $\mathbf{s}^{(0)} = \mathbf{x}_{t+1:t+h}$, and $\mathbf{s}^{(n)}$ is a noisy version of $\mathbf{s}^{(0)}$ (its level of corruption increases with $n$).

In the context of DYffusion's forecasting network, we can express its training objective (Eq. 3) in the following manner:

$$||F_\theta(\mathcal{I}_\phi(\mathbf{x}_t, \mathbf{x}_{t+h}, i_n), i_n) - \mathbf{x}_{t+h}||^2 = ||F_\theta(\mathcal{I}_\phi(\mathbf{x}_t, \mathbf{s}^{(N)}, i_n), i_n) - \mathbf{s}^{(N)}||^2 \tag{8}$$

$$= ||F_\theta(\mathbf{s}^{(n)}, i_n) - \mathbf{s}^{(N)}||^2, \tag{9}$$

where $\mathbf{s}^{(N)} = \mathbf{x}_{t+h}$ and $\mathbf{s}^{(n)} \approx \mathbf{x}_{t+i_n}$ is now a stepped backward in time version of $\mathbf{s}^{(N)}$. Note that the diffusion step indexing (superscript $n$) for DYffusion is reversed so that it aligns with the temporal indexing (subscript $t$), such that e.g. $\mathbf{s}^{(n)} \approx \mathbf{x}_{t+i_n}$ temporally precedes $\mathbf{s}^{(n+1)} \approx \mathbf{x}_{t+i_{n+1}}$. Accounting for the reversed indexing of the diffusion steps, it is easy to see the equivalence of the denoising network, $R_\theta$, in a conventional diffusion model, and its forecasting network counterpart, $F_\theta$, in DYffusion. The reason why DYffusion's forecaster network does not necessarily require the initial conditions, $\mathbf{x}_t$, as additional input is because the diffusion state, $\mathbf{s}^{(n)}$, depends on them. The central difference is the design and semantic meaning of the diffusion processes accompanying them. We visualize this in the diagrams in Fig. 1.

## C Diffusion process as ODE

In this section, we treat $\mathbf{x}$ as a continuous function of time, i.e. $\mathbf{x} : \mathbb{R} \to \mathbb{R}^n$, $s \mapsto \mathbf{x}(s)$. Here $s$ denotes the time variable (since $t$ is already used to define the index of temporal data snapshots).

During prediction, $\mathbf{x}_t$ is given, $\phi$ is fixed, and $\mathcal{I}_\phi$ only depends on $\hat{\mathbf{x}}_{t+h}$ and $i$. Therefore, we simplify notations by writing $\mathcal{I}_\phi(\mathbf{x}_t, \hat{\mathbf{x}}_{t+h}, i) = \mathcal{I}_{\phi, \mathbf{x}_t}(\hat{\mathbf{x}}_{t+h}, i)$. We will further omit the subscripts $\phi$ and $\mathbf{x}_t$.

## C.1 Cold Sampling from the Euler method

In this section, we show that Cold Sampling is an approximation of the Euler method for (5).

The Euler method for integrating $\mathbf{x}$ is

$$\mathbf{x}_{s+\Delta s} = \mathbf{x}_s + \Delta s \frac{d\mathcal{I}(F_\theta(\mathbf{x}_s, s), s)}{ds} \tag{10}$$

for a small $\Delta s$. We do not have access to $\frac{d\mathcal{I}(F_\theta(\mathbf{x}_s, s), s)}{ds}$. However, since we know $F_\theta$ and $\mathcal{I}_\phi$, we can approximate $\frac{d\mathcal{I}(F_\theta(\mathbf{x}_s, s), s)}{ds}$ by its first-order Taylor expansion around s:

$$\Delta s \frac{d\mathcal{I}(F_\theta(\mathbf{x}_s, s), s)}{ds} \approx \mathcal{I}(F_\theta(\mathbf{x}_{s+\Delta s}, s + \Delta s), s + \Delta s) - \mathcal{I}(F_\theta(\mathbf{x}_s, s), s) \tag{11}$$

This step can also be interpreted as evaluating the integral in (6) using the fundamental theorem of calculus. Then the Euler method becomes

$$\mathbf{x}_{s+\Delta s} = \mathbf{x}_s + \mathcal{I}(F_\theta(\mathbf{x}_{s+\Delta s}, s + \Delta s), s + \Delta s) - \mathcal{I}(F_\theta(\mathbf{x}_s, s), s) \tag{12}$$

Note that $\mathbf{x}_{s+\Delta s}$ on the right-hand side is unknown because it is the quantity we want to approximate in this step. A reasonable way to approximate $F_\theta(\mathbf{x}_{s+\Delta s}, s + \Delta s)$ is to replace it by $F_\theta(\mathbf{x}_s, s)$ because they both predict $\mathbf{x}(h)$ and use nearby points (assuming $\Delta s$ is small and $\mathbf{x}$ behaves nicely around $s$). The resulting update,

$$\mathbf{x}_{s+\Delta s} = \mathbf{x}_s + \mathcal{I}(F_\theta(\mathbf{x}_s, s), s + \Delta s) - \mathcal{I}(F_\theta(\mathbf{x}_s, s), s), \tag{13}$$

is exactly the Cold Sampling algorithm (Alg. 2). By formulating the diffusion process as an ODE, we have provided a new theoretical explanation for Cold Sampling.

Note that we made the approximation that $F_\theta(\mathbf{x}_{s+\Delta s}, s + \Delta s) \approx F_\theta(\mathbf{x}_s, s)$ to obtain the update rule (previous equation). The error introduced by this approximation is expected to be larger when $s$ is small. The intuition is as follows. When $s$ is small, the distance between $s$ and $t + h$ is large, and $F_\theta(\mathbf{x}_s, s)$ has to make a prediction farther ahead. Predicting far into the future is generally harder than predicting the near future. Therefore, the prediction error $F_\theta(\mathbf{x}_s, s) - \mathbf{x}_{t+h}$ is expected to be larger when $s$ is small. The larger uncertainty may lead to larger difference between $F_\theta(\mathbf{x}_{s+\Delta s}, s + \Delta s)$ and $F_\theta(\mathbf{x}_s, s)$.

To reduce the error brought by this approximation, it makes sense to sample more densely around the early part of the prediction window.

## C.2 Why is cold sampling better than naive sampling?

The cold sampling algorithm (adapted for DYffusion in Alg. 2) was proposed as an improved version over so-called "naive" sampling in the original Cold Sampling paper [2] for generalized diffusion model forward processes. Both

---

**Algorithm 4** Adapted Naive Sampling [2]

1: **Input:** Initial conditions $\hat{\mathbf{x}}_t := \mathbf{x}_t$, schedule $[i_n]_{i=0}^{N-1}$
2: **for** $n = 0, 1, \ldots, N - 1$ **do**
3:     $\hat{\mathbf{x}}_{t+h} \leftarrow F_\theta(\hat{\mathbf{x}}_{t+i_n}, i_n)$
4:     $\hat{\mathbf{x}}_{t+i_{n+1}} = \mathcal{I}_\phi(\mathbf{x}_t, \hat{\mathbf{x}}_{t+h}, i_{n+1})$  # difference w.r.t. Alg. 2
5: **end for**

---

sampling algorithms can be directly used to sample from DYffusion. For completeness, we adapt the simpler, naive sampling algorithm to our notation in Alg. 4, where the only difference to cold sampling (Alg. 2) lies in the missing error-correction terms in line 4 of the algorithms. In our ablation experiments, we find that cold sampling indeed outperforms naive sampling by a large margin (see *Naive sampling* row in Table 7), especially for the SST dataset where we use auxiliary diffusion steps. We explain by analyzing the discretization errors in the two sampling algorithms. In cold sampling, the discretization error per step is bounded by a term proportional to the step size $\Delta s$. Naive sampling does not have this property.

The true value of $\mathbf{x}$ at $s + \Delta s$ according to Equation (6) is

$$\begin{aligned}
\mathbf{x}(s + \Delta s) &= \mathbf{x}(s) + \int_s^{s+\Delta s} \frac{d\mathcal{I}_\phi(F_\theta(\mathbf{x}, s), s)}{ds} \\
&= \mathbf{x}(s) + \mathcal{I}(F_\theta(\mathbf{x}(s + \Delta s), s + \Delta s), s + \Delta s) - \mathcal{I}(F_\theta(\mathbf{x}(s), s), s). \tag{14}
\end{aligned}$$

Recall from Alg. 2 that given $\mathbf{x}(s)$, cold sampling predicts $\mathbf{x}(s + \Delta s)$ as

$$\hat{\mathbf{x}}(s + \Delta s) = \mathbf{x}(s) + \mathcal{I}_\phi(\mathbf{x}_{t-l:t}, F_\theta(\mathbf{x}(s), s), s + \Delta s) - \mathcal{I}_\phi(\mathbf{x}_{t-l:t}, F_\theta(\mathbf{x}(s), s), s). \quad (15)$$

The discretization error $e(\mathbf{x})$ of one step of cold sampling is the difference between the exact and predicted $\mathbf{x}(s + \Delta s)$:

$$
\begin{aligned}
e(\mathbf{x}, \Delta s) &= \mathbf{x}(s + \Delta s) - \hat{\mathbf{x}}(s + \Delta s) \\
&= \mathcal{I}(F_\theta(\mathbf{x}(s + \Delta s), s + \Delta s), s + \Delta s) - \mathcal{I}(F_\theta(\mathbf{x}(s), s), s + \Delta s). \quad (16)
\end{aligned}
$$

The following proposition states that $e(\mathbf{x})$ is bounded by a term proportional to the step size $\Delta s$.

**Proposition C.1.** *Assume that $F_\theta(\mathbf{x}(s), s)$ is Lipschitz in $s$. Assume also that $\mathcal{I}_\phi(\mathbf{x}_{t+h}, s)$ is Lipschitz in $\mathbf{x}_{t+h}$. The norm of the cold sampling discretization error, $||e(\mathbf{x}, \Delta s)||_2$, is bounded by $O(\Delta s)$.*

*Proof.* The proof relied on applying definitions of Lipschitz functions twice. Let $L_1$ be the Lipschitz constant for $F_\theta(\mathbf{x}(s), s)$ in $s$. Let $L_2$ be the Lipschitz constant for $\mathcal{I}(\mathbf{x}, s)$ in $\mathbf{x}$. Since $F_\theta(\mathbf{x}(s), s)$ is Lipschitz in $s$, we have $||F_\theta(\mathbf{x}(s + \Delta s), s + \Delta s) - F_\theta(\mathbf{x}(s), s)||_2 \leq L_1 \Delta s$. Since $\mathcal{I}(\mathbf{x}, s)$ is Lipschitz in $\mathbf{x}$, we have $||\mathcal{I}(F_\theta(\mathbf{x}(s + \Delta s), s + \Delta s), s + \Delta s) - \mathcal{I}(F_\theta(\mathbf{x}(s), s), s + \Delta s)||_2 \leq L_2 L_1 \Delta s$. Therefore $||e(\mathbf{x})|| \leq L_2 L_1 \Delta s$, which means the discretization error is bounded by a first-order term of the step size. $\square$

Under the same Lipschitz assumptions, the discretization error of the naive sampling is not guaranteed to be in the first order of step size. In naive sampling, the predicted $\mathbf{x}$ at time $s + \Delta s$ is

$$\hat{\mathbf{x}}(s + \Delta s) = \mathcal{I}(F_\theta(\mathbf{x}(s), s), s + \Delta s). \quad (17)$$

The discretization error of one step of naive sampling is $\mathbf{x}(s + \Delta s)$:

$$
\begin{aligned}
e(\mathbf{x}, \Delta s) =& \mathbf{x}(s + \Delta s) - \hat{\mathbf{x}}(s + \Delta s) \\
=& \mathcal{I}(F_\theta(\mathbf{x}(s + \Delta s), s + \Delta s), s + \Delta s) - \mathcal{I}(F_\theta(\mathbf{x}(s), s), s + \Delta s) \\
&+ \mathbf{x}(s) - \mathcal{I}(F_\theta(\mathbf{x}(s), s), s). \quad (18)
\end{aligned}
$$

Note that the first two terms are the same as the discretization error in cold sampling. However, the last two terms are not bounded by first-order terms of $\Delta s$. Hence, naive sampling can have larger discretization errors.

## D Experiments

### D.1 Datasets

#### D.1.1 SST data preprocessing

We create a new *sea surface temperatures* (SST) dataset based on NOAA OI SST V2 [30], which comes at a daily time scale. These data are available from 1982 to the present at a resolution of $1/4°$ degrees. For training, we use the years 1982-2018, for validation 2019, and for testing 2020. We have preprocessed the NOAA OI SST V2 dataset as follows:

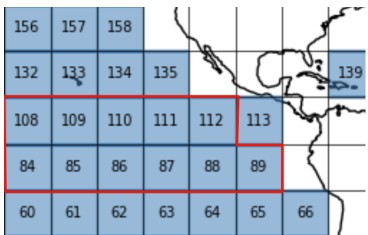

1. First, the globe is divided into $60 \times 60$ latitude $\times$ longitude grid tiles,

2. all tiles with less than $95\%$ of ocean cover are filtered out,

3. standardize the raw SSTs using daily means and standard deviations (computed on the training set only, i.e. 1982-2018),

4. replace continental NaNs with zeroes (after standardization), and

5. we subsample 11 grid tiles (covering mostly the eastern tropical Pacific, as shown in Fig. 5).

Figure 5: Visualization of the SST dataset that we created. It divides the globe into $60 \times 60$ latitude $\times$ longitude grid tiles. We only use the subset delineated in red, i.e. boxes 84-89 and 108-112.

Table 5: The hyperparameters used for each dataset. For the learning rates, we sweep over each value and report the best set of runs based on their validation CRPS computed on 50 samples. DYffusion $k$ refers to the number of artificial diffusion steps used, see Fig. 3. For architectural details, see D.3.

**Hyperparameters for each dataset**

| Hyperparameter | SST | Navier-Stokes | Spring Mesh |
|---|---|---|---|
| Batch size | 64 | 32 | 64 |
| Accumulate gradient batches | 4 | 2 | 1 |
| Max. Epochs | 50 | 200 | 300 |
| Gradient clipping (norm) | 1.0 | 1.0 | 1.0 |
| Learning rate(s) | 7e-4, 3e-4, 5e-5, 1e-5 | 7e-4, 3e-4 | 4e-4 |
| Weight decay | 1e-5 | 1e-4 | 1e-4 |
| AdamW $\beta_1$ | 0.9 | 0.9 | 0.9 |
| AdamW $\beta_2$ | 0.99 | 0.99 | 0.99 |
| DYffusion $k$ | 25 | 0 | 0 |

### D.1.2 Navier-Stokes and spring mesh datasets

We refer to the physical systems benchmark paper for more details regarding the Navier-Stokes and spring mesh benchmark datasets [44]. We follow the same evaluation splits and procedure (only complemented by the probabilistic skill metrics that we employ too). We always use the full training set provided by [44]. We use the Navier-Stokes dataset with four obstacles.

### D.2 Implementation Details

The set of hyperparameters that we use for each dataset, such as the learning rate and maximum number of epochs, can be found in Table 5. For all experiments, we use a floating point precision of 16 and do not use a learning rate scheduler. All diffusion models, including DYffusion, are trained with the L1 loss, while all barebone UNet/CNN networks are trained on the L2 loss. We use three different dropout rates for the SST UNet: 1) before the query-key-value projection of each attention layer, $dr_{at}$; 2) After the first sub-block of each ResNet block, $dr_{bl_1}$; 3) After the second sub-block of each ResNet block, $dr_{bl_2}$, where the first ResNet sub-block consists of convolution $\rightarrow$ normalization $\rightarrow$ time-embedding scale-shift $\rightarrow$ activation function, and the second sub-block is the same but without the time-embedding scale-shift.

**Perturbation baseline** We perturb the initial conditions, $\mathbf{x}_t$, with small amounts of Gaussian noise $\epsilon \sim \mathcal{N}(0, \sigma_\epsilon \mathbf{I})$. We found that $\sigma_\epsilon^* = 0.05$ gave the lowest CRPS scores among all variances that we tried, $\sigma_\epsilon \in \{0.01, 0.03, 0.05, 0.07, 0.1, 0.15, 0.2\}$. We note that choosing larger variances results in better SSR scores, but significantly lower CRPS and MSE scores. Inference dropout was disabled for this baseline variant.

**Dropout baseline** For this baseline, we enable the barebone model's dropout during both training and inference. For the SST dataset, similarly to the interpolator network of DYffusion, we found that using high dropout rates results in better performance. An explanation could be that the SST UNet has more capacity than the other backbone architectures. Concretely, the following SST UNet dropout rates resulted in the best performance: $dr_{at} = dr_{bl_2} = 0.6, dr_{bl_1} = 0.3$. For Navier-Stokes and spring mesh, there is only one dropout hyperparameter, and the corresponding best model uses $0.2$ and $0.05$ as dropout rate, respectively (selected from a sweep over $\{0.05, 0.1, 0.15, 0.2, 0.25, 0.3\}$).

**DDPM** We found that the cosine (linear) noise schedule gives better results for the SST (Navier-Stokes) dataset, and always uses the "predict noise" objective. For the SST dataset, the best performing DDPM is trained with 5 diffusion steps, while for Navier-Stokes it is trained with 500 steps. For Navier-Stokes, we found that while a DDPM with 5 or 10 diffusion steps can give good validation scores (or even better ones than the 500-step DDPM), it ends up diverging at test time after a few autoregressive iterations when used to forecast full trajectories.

**MCVD [67]** We train MCVD with 1000 diffusion steps for all datasets, as we were not able to successfully train it with fewer diffusion steps. We use a linear noise schedule (we found the cosine schedule to produce inferior results) using the "predict noise" objective. Due to the inference runtime

complexity of using 1000 diffusion steps, we only report one MCVD run in our main SST results. Note that in none of the experiments we use the UNet-based diffusion model backbone originally used by MCVD. In preliminary experiments on the SST dataset, however, we found that wrapping MCVD around the SST UNet resulted in slightly improved scores over wrapping it around the original MCVD UNet.

**DYffusion** For the SST dataset, we use 25 artificial diffusion steps (analogous to the schedule in green in Fig. 3), while for the Navier-Stokes and spring mesh datasets, we do not use any, i.e. $S = [j]_{j=0}^{h-1}$. Furthermore, we found that the refinement step of Alg. 2 did not improve performance for the SST dataset so we did not use it there, whereas it did improve performance for Navier-Stokes and spring mesh (see D.5.2). As for the choice of the interpolator network, $\mathcal{I}_\phi$, we conduct a sweep over the dropout rates for each dataset (analogous to the "Dropout" baseline). This is an important hyperparameter since we found that stochasticity in the interpolator is crucial for the overall performance of DYffusion. The interpolator network is selected based on the lowest validation CRPS. For the SST dataset, the selected $\mathcal{I}_\phi$ uses $dr_{at} = dr_{bl_2} = 0.6, dr_{bl_1} = 0$. The dropout rates for Navier-Stokes and spring mesh are $0.15$ and $0.05$, respectively. Generally, we found that the optimal amount of dropout for any given dataset strongly correlates between the "Dropout" multi-step forecasting baseline and DYffusion's interpolator network, $\mathcal{I}_\phi$. Thus, for a new dataset or problem, it is a valid strategy to sweep over the dropout rate for just one of the two model types. Motivated by the intuition that temporal interpolation is a simpler task than forecasting, the channel dimensionality of $\mathcal{I}_\phi$ is only 32 (instead of 64) for the first downsampling block of the SST UNet. To feed the two inputs $\mathbf{x}_t$ and $\mathbf{x}_{t+h}$ into the interpolator, we simply concatenate these snapshots along the channel dimensions, effectively doubling the number of input channels.

**Evaluation** All experiments are usually run over three random seeds when compute constraints allow. We use a 50-member ensemble for all reported results unless noted otherwise to compute the CRPS, mean squared error (MSE), and spread-skill ratio (SSR). The MSE is computed on the ensemble mean prediction. The SSR is defined as the ratio of the square root of the ensemble variance to the corresponding ensemble RMSE. It serves as a measure of the reliability of the ensemble, where values smaller than 1 indicate underdispersion (i.e. the probabilistic forecast is overconfident in its forecasts), and larger values overdispersion [16, 18]. We use the implementation in the `xskillscore`[4] Python package to compute the CRPS of the ensemble forecasts. The CRPS penalizes both the absolute skill as well as the sharpness of the ensemble of forecasts. It rewards small spread (i.e. sharpness) if the forecast is accurate, and measures the difference between the forecasted and observed cumulative distribution function (CDF). For a deterministic forecast, the CRPS reduces to the mean absolute error. For CRPS and MSE *lower is better*, for SSR *closer to 1 is better*.

### D.3  Neural Architecture Details

**SST UNet** For the SST dataset, we use a UNet implementation commonly used as backbone architecture of diffusion models[5]. The UNet for the SST dataset consists of three downsampling and three upsampling blocks. Each block consists of two convolutional residual blocks (ResNet blocks), followed by an attention layer, and a downsampling (or upsampling) module. Each ResNet block can be further divided into two sub-blocks. The first one consists of convolution → normalization → time-embedding scale-shift → activation function, and the second sub-block is the same but without the time-embedding scale-shift. The downsampling module is a 2D convolution that halves the spatial size of the input (with a $4 \times 4$ kernel and stride$= 2$). The upsampling module doubles the spatial size via nearest neighbor upsampling followed by a 2D convolution (with $3 \times 3$ kernel). At the end of each downsampling (upsampling) block the spatial size is halved (doubled) and the channel dimension is doubled (halved). We use 64 channels for the initial downsampling block, which means that the channel dimensionalities are $64 \rightarrow 128 \rightarrow 256$ and the spatial dimensions $(60, 60) \rightarrow (30, 30) \rightarrow (15, 15)$ in the corresponding downsampling blocks (reversed for the upsampling blocks). We use three different dropout rates for the SST UNet: 1) before the query-key-value projection of each attention layer, $dr_{at}$; 2) After the first sub-block of each ResNet block, $dr_{bl_1}$; 3) After the

---

[4]`https://xskillscore.readthedocs.io/`
[5]`https://github.com/lucidrains/denoising-diffusion-pytorch/blob/main/denoising_diffusion_pytorch/denoising_diffusion_pytorch.py`

Table 6: Navier-Stokes results, including on the out-of-distribution (OOD) test set. Left columns are same as in Table 1. As observed by Fig. 12 in [44], the differences across main vs OOD test set are minor.

| Method | Test | | | Out of Distribution | | |
|---|---|---|---|---|---|---|
| | CRPS | MSE | SSR | CRPS | MSE | SSR |
| Dropout | $0.078 \pm 0.001$ | $0.027 \pm 0.001$ | $0.715 \pm 0.005$ | $0.078 \pm 0.001$ | $0.027 \pm 0.001$ | $0.715 \pm 0.006$ |
| DDPM | $0.180 \pm 0.004$ | $0.105 \pm 0.010$ | $0.573 \pm 0.001$ | $0.189 \pm 0.012$ | $0.113 \pm 0.013$ | $0.555 \pm 0.023$ |
| MCVD | $0.154 \pm 0.043$ | $0.070 \pm 0.033$ | $0.524 \pm 0.064$ | $0.170 \pm 0.046$ | $0.082 \pm 0.036$ | $0.489 \pm 0.061$ |
| DYffusion | $\mathbf{0.067} \pm \mathbf{0.003}$ | $\mathbf{0.022} \pm \mathbf{0.002}$ | $\mathbf{0.877} \pm \mathbf{0.006}$ | $\mathbf{0.069} \pm \mathbf{0.002}$ | $\mathbf{0.023} \pm \mathbf{0.002}$ | $\mathbf{0.882} \pm \mathbf{0.002}$ |

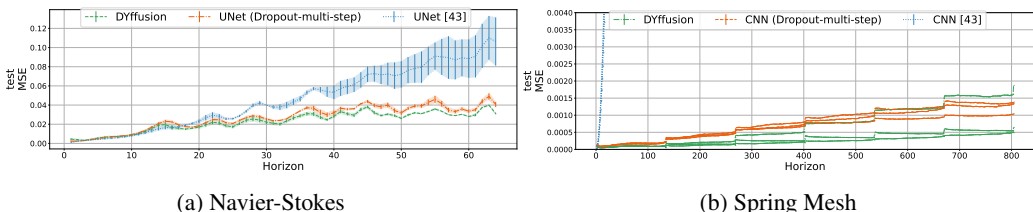

(a) Navier-Stokes        (b) Spring Mesh

Figure 6: Comparison against single-step deterministic baselines from [44]. We plot the MSE as a function of the rollout step (predicted horizon). It is clear that both our stochastic multi-step baseline, *Dropout-multi-step*, and DYffusion outperform the UNet (CNN) models from [44] on the Navier-Stokes (spring mesh) datasets of [44]. As reported in [44], the single-step CNN diverges after a few autoregressive steps on the spring mesh dataset. In addition, DYffusion performs especially well on long-range forecasts relative to the baselines. For spring mesh we plot the three forecasted rollout samples of each model separately due to higher variance between samples.

second sub-block of each ResNet block, $dr_{bl_2}$. For all models except the Dropout baseline and $\mathcal{I}_\phi$ in DYffusion, which use higher dropout rates, we use $dr_{at} = 0.1, dr_{bl_1} = 0, dr_{bl_2} = 0.3$.

**Navier-Stokes UNet and spring mesh CNN** For the Navier-Stokes and spring mesh benchmark datasets from [44], we simply re-use their proposed UNet and CNN architecture for the respective dataset. The only modification is the integration of the SST UNet time embedding module into the architectures of [44], as described below.

**Time embedding module** The time-embedding scale-shift, taken from the SST UNet, is a key component of all architectures since it enables them to condition on the diffusion step (for DDPM and MCVD) or the dynamical timestep (for the time-conditioned barebone models as well as for both $F_\theta$ and $\mathcal{I}_\phi$ in DYffusion). It is implemented by a sine and cosine-based featurization of the scalar diffusion step/dynamical timestep. These features are projected by a linear layer, followed by a GeLU activation, and another linear layer, which results in a "time embedding". Then, separately for each convolutional (or ResNet) block of the neural architecture the "time embedding" is further processed by a SiLU activation and another linear layer whose output is interpreted as two vectors that are used to scale and shift the block's inputs. In all architectures, the scale-shift operation is performed after the convolution and normalization layers, but before the activation function and dropout layer.

## D.4 Additional results

### D.4.1 Navier-Stokes out-of-distribution

In Table 6 we show the out-of-distribution test results for the Navier-Stokes datasets (analogously to Table 2 for the spring mesh dataset). As mentioned in the main text and observed by [44], the differences compared to the main test set are marginal.

### D.4.2 Benchmark against deterministic models

In this set of experiments, we directly compare our method against the deterministic single-step forecasting baselines presented in the benchmark dataset for the Navier-Stokes and spring mesh exper-

Table 7: DYffusion ablation. We change one component at a time in DYffusion, starting with all components enabled (first row). For SST, we perform the ablation only on a subset of the test dataset (box 88 in Fig. 5). For the Navier-Stokes and spring mesh datasets, we use the full test sets. The second to fifth rows are ablations at sampling/inference time: *No refinement* refers to not using the refinement step in line 6 of Alg. 2. *No $\mathcal{I}_\phi$ dropout* refers to disabling the inference dropout of the interpolator network, $\mathcal{I}_\phi$. *No $\mathcal{I}_\phi$ dropout & $\sigma_\epsilon$* refers to disabling the inference dropout of $\mathcal{I}_\phi$ and perturbing the inputs by $\sigma_\epsilon = 0.05$ (as for the Perturbation baseline). *Naive sampling* refers to swapping cold sampling (Alg. 2) with the naive sampling algorithm from [2]. The bottom three rows apply to both training and inference and ablate the choice of the forecaster conditioning $\mathbf{c}(\mathbf{x}_t, n)$, where for the last row $\alpha_n = \frac{n}{N-1}$. Here, the best choice varies across datasets.

| | SST | | | Navier-Stokes | | | Spring Mesh | | |
|---|---|---|---|---|---|---|---|---|---|
| | CRPS | MSE | SSR | CRPS | MSE | SSR | CRPS | MSE | SSR |
| Base | $0.181_{\pm 0.002}$ | $0.111_{\pm 0.002}$ | $1.04_{\pm 0.01}$ | $0.067_{\pm 0.003}$ | $0.022_{\pm 0.002}$ | $0.88_{\pm 0.01}$ | $0.0103_{\pm 0.0025}$ | $4.2\text{e-}4_{\pm 2.1\text{e-}4}$ | $1.13_{\pm 0.08}$ |
| No refinement | $0.181_{\pm 0.002}$ | $0.111_{\pm 0.002}$ | $1.08_{\pm 0.00}$ | $0.069_{\pm 0.003}$ | $0.024_{\pm 0.002}$ | $1.12_{\pm 0.02}$ | $0.0246_{\pm 0.0012}$ | $6.8\text{e-}4_{\pm 2.8\text{e-}4}$ | $2.05_{\pm 0.13}$ |
| No $\mathcal{I}_\phi$ dropout | $0.320_{\pm 0.009}$ | $0.206_{\pm 0.012}$ | $0.00_{\pm 0.00}$ | $0.098_{\pm 0.005}$ | $0.028_{\pm 0.003}$ | $0.00_{\pm 0.00}$ | $0.0337_{\pm 0.0038}$ | $2.5\text{e-}3_{\pm 6.4\text{e-}4}$ | $0.01_{\pm 0.00}$ |
| No $\mathcal{I}_\phi$ dropout & $\sigma_\epsilon$ | $0.308_{\pm 0.009}$ | $0.197_{\pm 0.012}$ | $0.40_{\pm 0.01}$ | $0.096_{\pm 0.005}$ | $0.028_{\pm 0.003}$ | $0.23_{\pm 0.00}$ | $0.0282_{\pm 0.0031}$ | $2.7\text{e-}3_{\pm 6.5\text{e-}4}$ | $0.98_{\pm 0.05}$ |
| Naive sampling | $0.681_{\pm 0.062}$ | $0.945_{\pm 0.117}$ | $0.52_{\pm 0.03}$ | $0.088_{\pm 0.004}$ | $0.029_{\pm 0.002}$ | $0.54_{\pm 0.01}$ | $0.0115_{\pm 0.0035}$ | $4.5\text{e-}4_{\pm 2.6\text{e-}4}$ | $0.76_{\pm 0.06}$ |
| $\mathbf{c} = \texttt{null}$ | $0.182_{\pm 0.002}$ | $0.111_{\pm 0.002}$ | $0.94_{\pm 0.02}$ | $0.067_{\pm 0.003}$ | $0.022_{\pm 0.002}$ | $0.88_{\pm 0.01}$ | $104_{\pm 111}$ | $9.9\text{e+}7_{\pm 1.4\text{e+}8}$ | $1.75_{\pm 0.16}$ |
| $\mathbf{c} = \mathbf{x}_t$ | $0.196_{\pm 0.002}$ | $0.128_{\pm 0.002}$ | $1.13_{\pm 0.01}$ | $0.077_{\pm 0.006}$ | $0.028_{\pm 0.003}$ | $0.77_{\pm 0.02}$ | $0.0103_{\pm 0.0022}$ | $4.2\text{e-}4_{\pm 2.1\text{e-}4}$ | $1.13_{\pm 0.08}$ |
| $\mathbf{c} = \alpha_n \mathbf{x}_t + (1-\alpha_n)\epsilon$ | $0.181_{\pm 0.002}$ | $0.111_{\pm 0.002}$ | $1.04_{\pm 0.01}$ | $0.074_{\pm 0.003}$ | $0.026_{\pm 0.001}$ | $0.83_{\pm 0.02}$ | $2.2044_{\pm 1.6889}$ | $3.6\text{e+}2_{\pm 3.5\text{e+}2}$ | $1.63_{\pm 0.04}$ |

iments [44] in terms of MSE. It is important to emphasize that our method is specifically designed to generate ensemble-based probabilistic forecasts, which is why our baselines and evaluation procedure focus on the probabilistic forecasting setting. Besides, MSE can only be part of the evaluation of a probabilistic forecasting model, since it does not capture the skill of the forecasted distribution like the CRPS or SSR metrics do. In Fig 6 it is visible that the baselines from [44] either degrade or diverge during the inference rollouts for the Navier-Stokes and spring mesh test evaluations, respectively. Both our method as well as our multi-step Dropout baseline outperform [44], especially in the long-range forecasting regime. This is likely because our models are trained to forecast multiple timesteps, while the models from [44] only learn to forecast the next timestep. As a result, the training objective significantly deviates from the evaluation procedure. The struggle with long-term stability and long-range forecasts by the benchmark datasets baselines was already noted by the dataset paper itself [44]. Note that architecturally all these models are almost identical, using the neural backbones proposed in [44].

### D.5 Ablations

#### D.5.1 Cold sampling versus naive sampling

As mentioned in section C.2, we find clear evidence that the DDIM-like cold sampling (Alg. 2) proposed by [2] for "generalized diffusion models" is very important to attain good samples from DYffusion. In Table 7, cold sampling corresponds to the first row, *Base*, and the *Naive sampling* row refers to swapping cold sampling (Alg. 2) with the naive sampling algorithm from [2] (see Alg. 4). The finding that naive sampling yields poor results for DYffusion and, more generally, "generalized diffusion models" is consistent with the findings from [2].

#### D.5.2 Inference dropout in the interpolator network

In Table 7 we show that disabling the inference dropout in the interpolator network, $\mathcal{I}_\phi$, results in considerably worse scores. This is to be expected, since without stochasticity in $\mathcal{I}_\phi$, our current framework collapses to generating deterministic forecasts (since the sampling algorithm and forecaster network are deterministic, and we assume that the given initial conditions are fixed). In such a case, computing the CRPS collapses to the mean absolute error, and the SSR becomes $0$ since there is no spread in the predictions. To attain an ensemble of forecasts, but keeping the interpolator dropout disabled, we also include an ablation row where we perturb the initial conditions with small random noise $\epsilon \sim \mathcal{N}(0, \sigma_\epsilon \mathbf{I})$, where we use $\sigma_\epsilon = 0.05$.

#### D.5.3 Refining the interpolation forecasts after sampling

We find in Table 7 (*No ref.* row), that the addition of line 6 to the cold sampling algorithm (see Alg. 2) can sometimes improve performance. However, this is not consistent across datasets: While for the

SST dataset, we hardly observe any difference, the scores degrade considerably when not refining the forecasts for the spring mesh dataset. A reason could be the relatively long training horizon used for spring mesh (134 for spring mesh versus 16 or 7 for Navier-Stokes or SST). In practice, we recommend that practitioners train DYffusion with the refinement step disabled to accelerate inference time (since the refinement step requires an additional forward pass per output timestep). Then, during evaluation it is encouraged to perform inference with DYffusion with both disabled and enabled refinement step to analyze whether enabling the refinement step can meaningfully improve the forecasts.

### D.5.4 Forecaster network conditioning

In the bottom three rows of Table 7 we ablate the three forecaster conditioning options (see B for detailed definitions), $c(x_t, n)$. The optimal choice of $c$ varies across datasets. In particular, for the spring mesh experiments, it turns out to be crucial to use the clean initial conditions (variant 2, where $c(x_t, \cdot) = x_t$) as conditioning (the two other variants diverge when forecasting the full test trajectories). A likely explanation for this is that the training horizon of $h = 134$ is relatively long for spring mesh. As a result, the forecaster network can greatly benefit from being explicitly conditioned on the initial conditions during all diffusion states (as opposed to only being indirectly influenced by $x_t$ through the interpolator network). In practice and if compute allows, it may be worthwhile to train DYffusion with at least the first two variants for a few iterations to analyze if any conditioning variant strongly outperforms the others.

### D.5.5 Choosing the training horizon

In any multi-step forecasting model, the training horizon, $h$, is a key hyperparameter choice. Usually, its choice is constrained by the number of timesteps that fit into GPU memory, and it is expected that larger training horizons will improve performance when evaluated on long (autoregressive) rollouts. However, for continuous-time models including ours, where the number of timesteps needed in GPU memory does not change as a function of $h$ (see Table 10), the choice of $h$ is flexible. In Table 8, we explore using three different training hori-

Table 8: Navier-Stokes ablation of the training horizon, $h$. All methods are evaluated on the 64-step test trajectories. For example, for $h = 1$ ($h = 16$) the corresponding methods are unrolled autoregressively 64 (4) times.

| Method | CRPS | MSE | SSR |
|---|---|---|---|
| Dropout ($h = 1$) | $0.132 \pm 0.006$ | $0.046 \pm 0.006$ | $0.002 \pm 0.00$ |
| Dropout ($h = 8$) | $0.086 \pm 0.012$ | $0.026 \pm 0.002$ | $0.416 \pm 0.29$ |
| Dropout ($h = 16$) | $0.078 \pm 0.001$ | $0.027 \pm 0.001$ | $0.715 \pm 0.01$ |
| Dropout ($h = 32$) | $0.078 \pm 0.001$ | $0.025 \pm 0.001$ | $0.651 \pm 0.01$ |
| DYffusion ($h = 8$) | $0.076 \pm 0.002$ | $0.027 \pm 0.001$ | $0.701 \pm 0.02$ |
| DYffusion ($h = 16$) | $\mathbf{0.067} \pm \mathbf{0.003}$ | $\mathbf{0.022} \pm \mathbf{0.002}$ | $\mathbf{0.877} \pm \mathbf{0.01}$ |
| DYffusion ($h = 32$) | $0.075 \pm 0.003$ | $0.028 \pm 0.001$ | $0.862 \pm 0.04$ |

zons ($h \in \{8, 16, 32\}$) for the Navier-Stokes dataset for both the barebone time-conditioned Dropout model as well as DYffusion. For DYffusion, this means that we train both an interpolator network as well as a corresponding forecaster net with the same training horizon. Note that $h = 16$ corresponds to the main results, and is a sweet spot for DYffusion for this task. However, any of the used horizons results in better scores than the best baseline (for any baseline training horizon). We also include a next-step forecasting baseline ($h = 1$), which results in significantly worse results than any multi-step forecasting approach. This is likely due to the compounding effect of autoregressive errors.

### D.5.6 Using auxiliary diffusion steps can improve performance

In Table 9 we ablate the number of artificial diffusion steps, $k$, used to train/evaluate DYffusion on the SST dataset. As in Table 7 we evaluate on the test box 88 only. It is clear, that $k > 0$ is important for optimal CRPS and SSR performance. Here, the $k$ auxiliary diffusion steps correspond to the floating-point interpolation timesteps $\{\frac{j}{k+1}\}_{j=1}^{k}$, as visualized for a toy example in Fig. 9b. Interestingly, we did not observe this phenomenon for the Navier-Stokes or spring mesh datasets, where using $k > 0$ did not improve performance over using the simplest schedule $S = [i_n]_{n=0}^{N-1} = [j]_{j=0}^{h-1}$, where the number of diffusion steps and the horizon are equal ($k = 0$).

### D.5.7 Forecaster network iteratively refines its forecasts

In this set of experiments, we verify that the forecaster's prediction of $x_{t+h}$ for a given diffusion step $n$ and associated input $\hat{x}_{t+i_n}$ usually improves as $n$ increases, i.e. as we approach the end of

Table 9: For the SST dataset, using $k$ extra artificial diffusion steps corresponding to floating-point $i_k$, especially such that $i_k < 1$ for all $k$ (similarly to Fig. 9b), improved performance significantly. The simplest possible schedule with $k = 0$ corresponds to $S = [i_n]_{n=0}^{N-1} = [j]_{j=0}^{h-1}$, where the number of diffusion steps and the horizon are equal. The CRPS does not vary significantly when $k > 0$, while the SSR increases steadily with larger $k$. This can be explained by larger $k$ creating a longer sampling sequence with $k$ extra diffusion steps that may lead to more diverse samples. We evaluate only on a subset of the test dataset (box 88 in Fig. 5).

| $k$ | CRPS | MSE | SSR |
|---|---|---|---|
| $k = 0$ | $0.208 \pm 0.002$ | $0.107 \pm 0.001$ | $0.49 \pm 0.00$ |
| $k = 10$ | $0.184 \pm 0.002$ | $0.113 \pm 0.003$ | $0.84 \pm 0.02$ |
| $k = 25$ | $0.181 \pm 0.001$ | $0.111 \pm 0.001$ | $0.99 \pm 0.02$ |
| $k = 40$ | $0.184 \pm 0.002$ | $0.113 \pm 0.002$ | $1.06 \pm 0.03$ |
| $k = 45$ | $0.189 \pm 0.004$ | $0.119 \pm 0.005$ | $1.09 \pm 0.03$ |

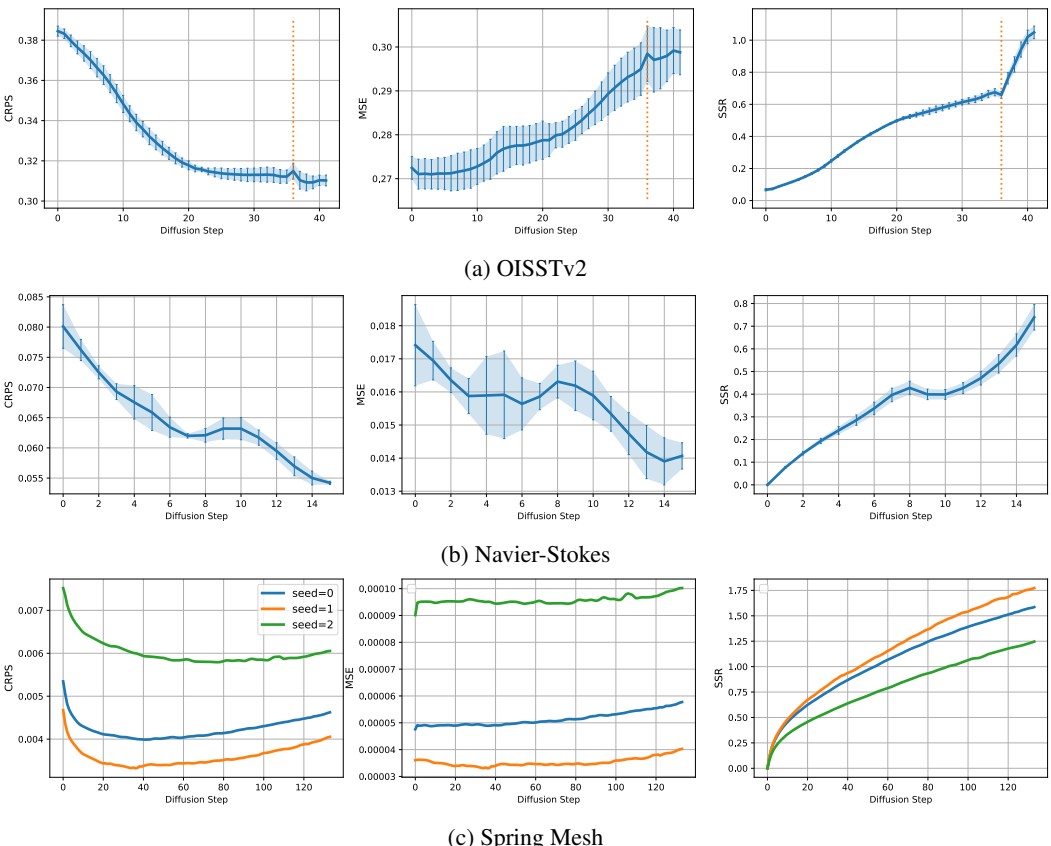

(a) OISSTv2

(b) Navier-Stokes

(c) Spring Mesh

Figure 7: We verify that the forecasts $\hat{\mathbf{x}}_{t+h} = F_\theta(\hat{\mathbf{x}}_{t+i_n}, i_n)$ of the forecaster network (i.e. the diffusion model backbone) usually improve as the reverse process comes closer in time to $t + h$ (i.e. as $n$ and $i_n$ increase) in terms of CRPS and spread-skill ratio (SSR). Interestingly, the MSE scores tend to degrade for the SST and spring mesh datasets (7a and 7c). For the SST dataset (7a), we also visualize a dotted vertical, orange line at diffusion step 36, which corresponds to the end of the 35 auxiliary diffusion steps (i.e. with floating-point $i_n < 1$), which are followed by the diffusion steps corresponding to the data resolution (i.e. $i_n \in \{1, 2, \ldots, 6\}$). The SSR score improves significantly after that diffusion step. To achieve a clearer visualization of the spring mesh dataset, where the variance across random seeds is relatively large, we plot each run separately. Interestingly, for spring mesh, the CRPS starts degrading after around 30 to 70 diffusion steps, which may indicate that a shorter training horizon could be more amenable for performance.

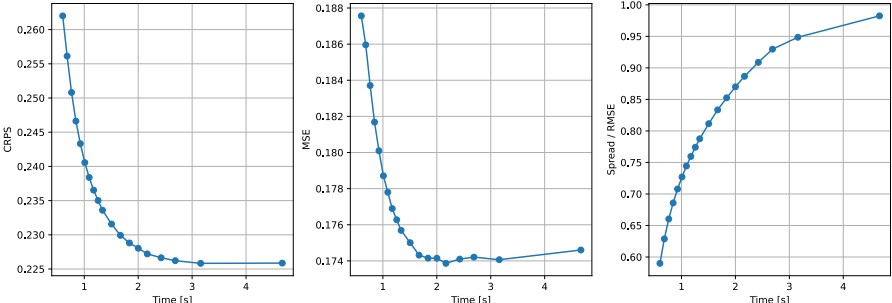

Figure 8: There is a clear trade-off between inference speed (x-axis) and performance (y-axis) as a function of the number of diffusion steps used for inference by DYffusion. Here, we show the SST test scores for one run of DYffusion, which was trained with $35$ auxiliary diffusion steps (on top of the $7$ given by the data). Each dot from left to right represents performing inference with an increasing number of diffusion steps. Each dot uses the base schedule $S_{base} = \{0, 1, \ldots, h - 1\}$, where $h = 7$, plus $N_{aux}$ additional diffusion steps drawn from the ones used for training. $N_{aux} = (1, 2, 3, 4, 5, 6, 7, 8, 9, 10, 12, 14, 16, 18, 20, 23, 26, 29, 35)$, and each such additional diffusion step corresponds to an implicit dynamical timestep in $(0, 1)$. Interestingly, almost equivalent (for CRPS) or slightly better (for MSE) scores can be sometimes obtained by using fewer diffusion steps than used for training (right-most dots of each subplot), which immediately benefits inference speed.

the reverse process and $\mathbf{x}_{t+i_n}$ gets closer in time to $\mathbf{x}_{t+h}$ (see Fig. 7). This analysis is conducted on a subset of the SST test set (box 112 in Fig. 5) and validation sets of Navier-Stokes and spring mesh, for the respective datasets, using a 20-member ensemble. We use the validation sets of the physical systems benchmark because the scope of the analysis only extends up to the training horizon (as opposed to the full test set trajectories).

### D.5.8 Accelerated DYffusion sampling

In Fig. 8 we study how sampling from DYffusion can be accelerated in a similar way to how DDIM [61] can accelerate sampling from Gaussian diffusion models. The continuous-time nature of the backbone networks in DYffusion invites the use of arbitrary dynamical timesteps as diffusion states during inference. Thus, to accelerate sampling, we can skip some of the $N$ diffusion steps used for training, $S_{train} = \{i_n\}_{n=0}^{N-1}$, which automatically results in fewer neural network forward passes. For example, we can only use the base schedule $S_{base} = \{0, 1, \ldots, h - 1\}$, where the diffusion states correspond to the temporal resolution of the dynamical data in a one-to-one mapping (see black arrows in Fig. 3). In Fig. 8, we start with $S_{base}$ as inference schedule (left-most dots in each subplot) for the SST dataset, and then incrementally add more diffusion steps from $S_{train} \setminus S_{base}$ to it, until reaching the full training schedule (right-most dots). We find that sampling can be significantly accelerated with marginal drops in CRPS and MSE performance. Note that the dynamical timesteps needed as outputs of DYffusion or for downstream applications pose a lower bound (here, $S_{base}$) on how much we can accelerate our method since any such output timestep needs to be included in the sampling schedule or in the set of output timesteps, $J$ (line 6 in Alg. 2). For that reason, it is not straightforward to accelerate the Navier-Stokes and spring mesh DYffusion runs, since their training schedules already correspond to $S_{base}$.

### D.6 Modeling Complexity of DYffusion and Baselines

In Table 10 we enumerate the different modeling and computing requirements needed for each of the baselines and our method. Dropout (multi-step) refers to the barebone backbone network that forecasts all $h$ timesteps $\mathbf{x}_{t+1:t+h}$ in a single forward pass. Dropout (continuous) refers to the barebone backbone network that forecasts one timestep $\mathbf{x}_{t+k}$ for $k \in (1, h)$ in a single forward pass, conditioned on the time, $k$. Both methods perform similarly in our exploratory experiments (not shown), and in our experiments, we always report the scores of the time-conditioned (i.e. continuous) variant. The multi-step approach corresponds to the way the backbone model of a video diffusion model operates, while the continuous variant is similar to how the forecaster network in DYffusion

Table 10: We report the requirements needed to train a method to forecast up to $h$ steps into the future, where $c$ refers to the number of input/output channels (e.g. 1 for SST data), and $w$ to the window size (here, $w = 1$ for all experiments). In the second and third columns, we report the input and output channel dimensions, respectively, that the (backbone) neural network needs to have (assuming that the window dimension is concatenated to the channel dimension). $|\text{Mem}(\mathbf{x}_t)|$ refers to the number of timesteps that need to be present in (GPU) memory to compute the training objective. The last column denotes how many network forward passes are needed to get the forecasts for all $h$ timesteps. Here, $N_1, N_2$ refers to the number of (sampling) diffusion steps used by DDPM/MCVD and DYffusion, respectively. Usually, $N_1 > N_2 \geq h$, since Gaussian noise diffusion models will require more diffusion steps to attain comparable predictive skill. For Navier-Stokes and spring mesh $N_2 = h$, for SST $N_2 = 35$. For MCVD, $N_1 = 1000$. The factor of 3 for DYffusion is a result of the two extra interpolator network forward passes needed in line 4 of Alg. 2. For large horizons, the model size and memory requirements of multi-step models and conventional diffusion models can be prohibitive. It is clear that (video) diffusion models do not scale well for long horizons.

**Modeling complexity**

| Method | $c_{in}$ | $c_{out}$ | $|\text{Mem}(\mathbf{x}_t)|$ | #Forward |
|---|---|---|---|---|
| Dropout (continuous) | $w * c$ | $c$ | $w + 1$ | $h$ |
| Dropout (multi-step) | $w * c$ | $h * c$ | $w + h$ | $1$ |
| DDPM / MCVD | $(h + w) * c$ | $h * c$ | $w + h$ | $N_1$ |
| DYffusion | $w * c$ | $c$ | $w + 1$ | $3 * N_2$ |

operates. Video diffusion models have higher modeling complexity because they need to model the full "videos", $\mathbf{x}_{t+1:t+h}$, at each diffusion state (or corrupted versions of it). Especially for long horizons, $h$, and high-dimensional data with several channels, this complicates the learning task for the neural network. Meanwhile, our method is only slightly impacted by the choice of $h$.

## D.7 Example Sampling Trajectories Visualized

Example sampling trajectories of our approach, as a function of the schedule $i_n$, are visualized in Fig. 9, where the top one corresponds to the simplest case where we use a one-to-one mapping between diffusion steps, $n$, and interpolation/dynamical timesteps, $i_n$. Each of the intermediate $\hat{\mathbf{x}}_i$ can be used as a forecast for timestep $i$. The forecaster network, $F_\theta$, repeatedly forecasts $\mathbf{x}_h$, but does so with increasing levels of skill (analogously to how conventional diffusion models iteratively denoise/refine the predictions of the "clean" data, $\mathbf{s}^{(0)}$; see Fig. 7).

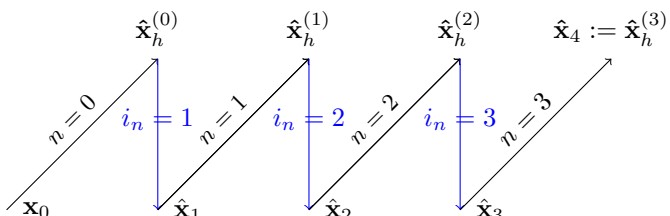

(a) Basic schedule, 1-to-1 diffusion step to dynamical timestep mapping

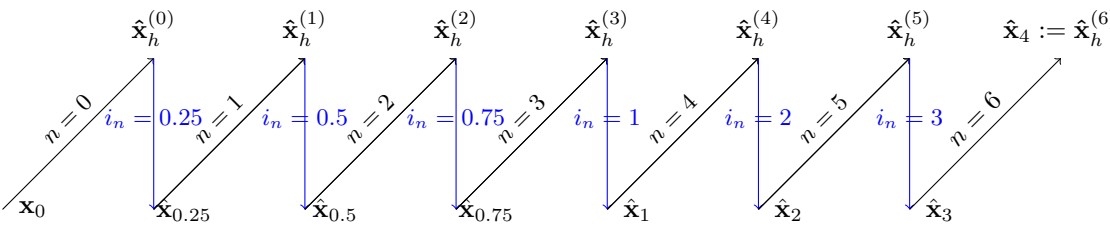

(b) Auxiliary, additional diffusion steps (here, $k = 3$) map uniformly to $(0, 1)$

Figure 9: Exemplary sampling trajectories of two different schedules for mapping diffusion steps, $n$, to interpolation timesteps, $i_n$. For convenience, Fig. 9a is a copy of Fig. 2. The schedules are illustrated using a horizon of $h = 4$. The **black** lines represent forecasts performed by the forecaster network, $F_\theta$. The first forecast is performed based on the initial conditions, $\mathbf{x}_0$. The **blue** lines represent the subsequent temporal interpolations performed by the interpolator network, $\mathcal{I}_\phi$.

