## A Appendix

## B Diffusion process as ODE

In this section, we treat $\mathbf{x}$ as a continuous function of time, i.e. let $\mathbf{x} : \mathbb{R} \to \mathbb{R}^n$, $s \mapsto \mathbf{x}(s)$. Here $s$ denotes the time variable (since $t$ is already taken and $n$ usually refers to discrete variables).

During prediction, $\mathbf{x}_t$ is given, $\phi$ is fixed, and $\mathcal{I}_\phi$ only depends on $\hat{\mathbf{x}}_{t+h}$ and $i$. Therefore, we simplify notations by writing $\mathcal{I}_\phi(\mathbf{x}_t, \hat{\mathbf{x}}_{t+h}, i) = \mathcal{I}_{\phi, \mathbf{x}_t}(\hat{\mathbf{x}}_{t+h}, i)$. We will further omit the subscripts $\phi$ and $\mathbf{x}_t$.

### B.1 Cold Sampling from the Euler method

In this section, we show that Cold Sampling is an approximation of the Euler method for (5).

The Euler method for integrating $\mathbf{x}$ is

$$\mathbf{x}_{s+\Delta s} = \mathbf{x}_s + \Delta s \frac{d\mathcal{I}(F_\theta(\mathbf{x}_s, s), s)}{ds} \tag{7}$$

for a small $\Delta s$. We do not have access to $\frac{d\mathcal{I}(F_\theta(\mathbf{x}_s, s), s)}{ds}$. However, since we know $F_\theta$ and $\mathcal{I}_\phi$, we can approximate $\frac{d\mathcal{I}(F_\theta(\mathbf{x}_s, s), s)}{ds}$ by its first-order Taylor expansion around s:

$$\Delta s \frac{d\mathcal{I}(F_\theta(\mathbf{x}_s, s), s)}{ds} \approx \mathcal{I}(F_\theta(\mathbf{x}_{s+\Delta s}, s + \Delta s), s + \Delta s) - \mathcal{I}(F_\theta(\mathbf{x}_s, s), s) \tag{8}$$

This step can also be interpreted as evaluating the integral in (6) using the fundamental theorem of calculus. Then the Euler method becomes

$$\mathbf{x}_{s+\Delta s} = \mathbf{x}_s + \mathcal{I}(F_\theta(\mathbf{x}_{s+\Delta s}, s + \Delta s), s + \Delta s) - \mathcal{I}(F_\theta(\mathbf{x}_s, s), s) \tag{9}$$

Note that $\mathbf{x}_{s+\Delta s}$ on the right hand side is unknown, because it is the quantity we want to approximate in this step. A reasonable way to approximate $F_\theta(\mathbf{x}_{s+\Delta s}, s + \Delta s)$ is to replace it by $F_\theta(\mathbf{x}_s, s)$, because they both predict $\mathbf{x}(h)$ and use nearby points (assuming $\Delta s$ is small and $\mathbf{x}$ behaves nicely around $s$). The resulting update,

$$\mathbf{x}_{s+\Delta s} = \mathbf{x}_s + \mathcal{I}(F_\theta(\mathbf{x}_s, s), s + \Delta s) - \mathcal{I}(F_\theta(\mathbf{x}_s, s), s), \tag{10}$$

is exactly the Cold Sampling algorithm (Alg. 2). By formulating the diffusion process as an ODE, we have provided a new theoretical explanation for Cold Sampling.

Note that we made the approximation that $F_\theta(\mathbf{x}_{s+\Delta s}, s + \Delta s) \approx F_\theta(\mathbf{x}_s, s)$ to obtain the update rule (previous equation). The error introduced by this approximation is expected to be larger when $s$ is small. The intuition is as follows. When $s$ is small, the distance between $s$ and $t + h$ is large, and $F_\theta(\mathbf{x}_s, s)$ has to make a prediction farther ahead. Predicting far into the further is generally harder than predicting the near future. Therefore, the prediction error $F_\theta(\mathbf{x}_s, s) - \mathbf{x}_{t+h}$ is expected to be larger when $s$ is small. The larger uncertainty may lead to larger difference between $F_\theta(\mathbf{x}_{s+\Delta s}, s + \Delta s)$ and $F_\theta(\mathbf{x}_s, s)$.

To reduce the error brought by this approximation, it makes sense to sample more densely around the early part of the prediction window.

### B.2 Why is cold sampling better than naive sampling?

In experiments, and consistent with prior work [2], cold sampling outperforms naive sampling by a large margin. We provide an explanation

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

$ (only implicitly through $h$ being some kind of lower bound on the number of diffusion steps in DYffusion).

## B.6 Sampling Trajectories

Example sampling trajectories of our approach, as a function of the schedule $i_n$, are visualized in Fig. 6, where the top one corresponds to the simplest case where we use a one-to-one mapping between diffusion steps, $n$, and interpolation/dynamical timesteps, $i_n$. Each of the intermediate $\hat{\mathbf{x}}_i$ can be used as a forecast for timestep $i$. The forecaster network, $F_\theta$, repeatedly forecasts $\mathbf{x}_h$, but does so with increasing levels of skill (analogously to how conventional diffusion models iteratively denoise/refine the predictions of the "clean" data, $\mathbf{s}^{(0)}$).

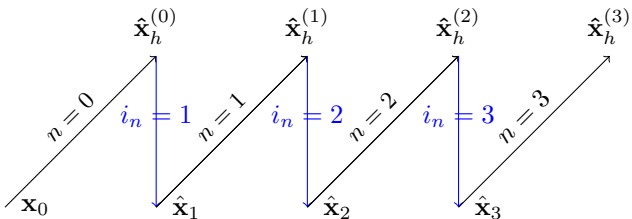

(a) Basic schedule, 1-to-1 diffusion step to dynamical timestep mapping

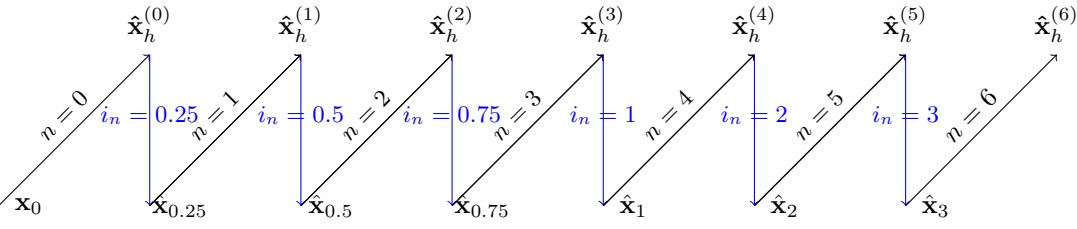

(b) Additional diffusion steps (here, 3) map uniformly to $(0, 1)$

Figure 6: Exemplary sampling trajectories of two different schedules for mapping diffusion steps, $n$, to interpolation (dynamical) timesteps, $i_n$. The schedules are illustrated using a horizon of $h = 4$. The black lines represent forecasts performed by the forecaster network, $F_\theta$. The first forecast is performed based on the initial conditions, $\mathbf{x}_0$. The blue lines represent the subsequent temporal interpolation performed by the interpolator network, $\mathcal{I}_\phi$.