# OpenReview forum: "DYffusion: A Dynamics-informed Diffusion Model for Spatiotemporal Forecasting"
_NeurIPS.cc/2023/Conference — NeurIPS 2023 poster_

### Official Review · Reviewer_4TEk · 2023-06-28

**Soundness:** 2 fair
**Presentation:** 2 fair
**Contribution:** 2 fair
**Rating:** 4
**Confidence:** 4

**Summary:**

This paper proposes DYffusion, an approach that mimics the diffusion models for spatiotemporal forecasting.
It treats the noising process as interpolation (parameterized by $I_\phi$) and the denoising process as forecasting (parameterized by $F_\theta$), i.e., it reimages the noising step $T$ in the original diffusion models as the temporal step in forecasting.
DYffusion parameterizes the probabilistic transitions in forecasting continuous-time trajectories, which is similar to parameterizing the diffusion of SDEs.
It performs competitively on spatiotemporal forecasting tasks, including sea surface temperatures, Navier-Stokes flows, and spring mesh systems, in terms of probabilistic skill score metrics.

**Strengths:**

1. DYffusion reimagines the continuous-time probabilistic forecasting problem as a diffusion process. It benefits from existing diffusion algorithms and methods to accelerate the inference sampling.
2. DYffusion achieves the best or competitive scores while significantly reducing the time cost compared to diffusion approaches.

**Weaknesses:**

1. Eq. (6) is incorrect. There should be a differential instead of a derivative in the integral.
2. DYffusion's method is closely related to neural ODE and SDE. However, empirical studies lack corresponding baselines. For example, one possible baseline could involve extrapolating $dF_\theta/ds$ using an ODE solver.
3. It is unfair to use $F_\theta$ trained with Algorithm 1 in the baseline method Dropout. The forecaster used in Dropout should be trained with an objective such as $||F_\theta(x_{t+i}, i)-x_{t+h}||^2$ instead.
4. One of the main contributions mentioned is that DYffusion reduces complexity. However, a complexity analysis is missing.
5. Neural SDE [1,2,3,4] parameterizes the stochastic dynamics for modeling continuous-time processes and is therefore inherently suitable for probabilistic continuous-time forecasting. However, there is no discussion on a range of related works on neural SDE.

[1] Deng, Ruizhi, et al. "Modeling continuous stochastic processes with dynamic normalizing flows." Advances in Neural Information Processing Systems 33 (2020): 7805-7815.

[2] Liu, Xuanqing, et al. "Neural sde: Stabilizing neural ode networks with stochastic noise." arXiv preprint arXiv:1906.02355 (2019).

[3] Jia, Junteng, and Austin R. Benson. "Neural jump stochastic differential equations." Advances in Neural Information Processing Systems 32 (2019).

[4] Kidger, Patrick, et al. "Efficient and accurate gradients for neural SDEs." Advances in Neural Information Processing Systems 34 (2021): 18747-18761.

**Questions:**

Why does Dropout perform so poorly at $t=2$ in Figure 3? Does this observation indicate the mentioned Weakness 3?

**Limitations:**

This paper includes a separate section for discussing limitations. There is no potential negative societal impact.

---

> ### Author Rebuttal · Authors · 2023-08-10
>
> Thank you  for your positive comments regarding the novelty, connection to existing diffusion models, and efficiency of our approach, as well as your valuable feedback to which we respond below.
>
> **Q1:**
> > Eq. (6) is incorrect. There should be a differential instead of a derivative in the integral.
>
> **A1:**
> Yes, there should be a $ds$ at the end of the equation. We apologize for the confusion, and have fixed this for the revised version.
>
> **Q2:**
> > DYffusion's method is closely related to neural ODE and SDE. However, empirical studies lack corresponding baselines. For example, one possible baseline could involve extrapolating dF_\theta/ds  using an ODE solver
>
> **A2:**
>
> Thank you for making the connection of DYffusion to neural ODE/SDE methods.
>
> However, **DYffusion is not directly related to neural SDEs**. They are different types of deep generative models (DGM). DYffusion is a diffusion model-based DGM that leverages the theory of SDE to learn high-dimensional distributions via score matching. In contrast, neural ODE/SDE is a type of autoregressive DGM that assumes the hidden states of a neural network to follow a particular ODE/SDE dynamics. Despite the shared use of SDE in these two lines of works, they have very different learning mechanisms and applications.
>
>  To the best of our knowledge, existing neural SDE methods (including your references [1-4]) only focus on low-dimensional problems. For example, the maximum dimensionality in [1] and [4] is four, respectively two, using an air quality dataset. Meanwhile, [2, 3] do not study dynamics forecasting: [2] only has image experiments, and [3] studies event prediction of low dimensional datasets. In our work, the spring mesh dataset with 400 = 10 x 10 x 4 has lowest dimensionality among all datasets, and we note that this dimensionality can further increase when performing multi-step forecasting.
>
> The reason for why these neural SDE papers only experiment with low-dimensional multivariate timeseries is exactly because of the ODE/SDE assumption on the hidden state mentioned above. Due to the need of solving a ODE/SDE through a numerical solver, neural ODE/SDE currently struggle with high-dimensional data such as video data in our work.  We would be more than willing to add any baselines that have been applied to high-dimensional spatiotemporal data similar to the one we study in this paper. We would appreciate any such pointers to potential baselines.
>
> **Q3:**
>
> > It is unfair to use $F_\theta$ trained with Algorithm 1 in the baseline method Dropout. The forecaster used in Dropout should be trained with an objective such as $||F_\theta(x_{t+i},i)−x_{t+ℎ}||^2$ instead.
>
>
> **A3:**
> The Dropout baseline was NOT trained with Alg. 1 (that one was only designed/used for DYffusion). Instead it was trained on the objective $||F_\theta(x_t, i) - x_{t+i}||^2$ for $1\leq i \leq h$. We realize that lines 237-238 and 242-244 may be confusing in this respect, and we will improve their clarity in our revised version. We will add the aforementioned objective that we used to train the Dropout baseline to the appendix (and refer to it in the main text). When we say that the baseline is trained _"analogously to the DYffusion forecaster"_ (line 243-44) we meant to only refer to the fact that both methods rely on a time-conditioning mechanism.
>
> **Q4:**
>
> > One of the main contributions mentioned is that DYffusion reduces complexity. However, a complexity analysis is missing.
>
> **A4:**
> See table 7 in the appendix for a comparison on how DYffusion effectively reduces the neural network input and output dimensionality, memory needs, and efficiency compared to a video diffusion model/MCVD.
>
> **Q5:**
>
> > Neural SDE [1,2,3,4] parameterizes the stochastic dynamics for modeling continuous-time processes and is therefore inherently suitable for probabilistic continuous-time forecasting. However, there is no discussion on a range of related works on neural SDE.
>
> **A5:**
> Thank you for the references. We will make sure to discuss and cite them, as well as the neural SDE literature more generally, in our revised version. As mentioned in **A2**, at this moment we don't believe that neural SDE methods are an appropriate baseline for the high-dimensional spatiotemporal forecasting problem studied in our paper, since all Neural SDE papers that we are aware of (incl. [1-4]) only cover low dimensional time series.
>
> **Q6:** _Figure 3_
>
> **A6:**
> > Why does Dropout perform so poorly at t=2 in Figure 3?
>
> That is a good observation. Generally, the beginning of the Navier-Stokes trajectories look qualitatively different to following timesteps where the fluid is already clearly "flowing". As such it is possible that those initial timesteps are harder to qualitatively forecast well. Plus, by definition, there are only relatively few training examples of the very start of the training trajectories compared to mid/end points of them. This might be especially a problem for underdispersive models (SSR < 1) that do not capture the full data distribution well.
>
> > Does this observation indicate the mentioned Weakness 3?
>
> No, see our answer **A3**.

---

> > ### Comment · Reviewer_4TEk · 2023-08-15
> >
> > Thank you for the clarification on the differences between DYffusion and neural ODE/SDE that I had overlooked. Based on your helpful explanation, I have some additional questions to better understand your work:
> >
> > Given that the horizon $h$ remains fixed once chosen, and considering that the training of the interpolator $\mathcal{I}$ is independent of the forecaster $\mathcal{F}$, does this imply that alternative forecasting models capable of predicting $x_{t+h}$ from $x_t$ can also be employed to achieve continuous forecasting in conjunction with the trained interpolator $\mathcal{I}$? If such is the case, it would be beneficial to conduct further ablation studies to elucidate the **necessity and advantages of the forecaster** $\mathcal{F}$. These studies could shed light on how the designed diffusion training and inference contribute to the efficacy of the forecaster.
> > For example, **exploring diffusion settings in common practice** could be insightful, i.e., investigating a forecaster that generates $x_{t+h}$ by denosing from Gaussian noise conditioned on $x_t$ might be worth considering, especially given the lack of evidence indicating that the intermediate $\hat{x}_{t+i_n}$s are utilized. It would be valuable to address why such a configuration was not selected and how it contrasts with the chosen approach.
> >
> > As for neural SDE on high-dimensional data,I would like to suggest considering Neural-SPDE [5], which applies neural SDE to Navier-Stokes equations in a 64x64 grid setting. Its codebase provides implementations of Neural-SPDE as well as various baselines, including NCDE and NRDE. I recommend leveraging these baselines to showcase the effectiveness of DYffusion against neural ODE/SDE based approaches, if time permits for an empirical comparison.
> >
> > [5] Neural Stochastic PDEs: Resolution-Invariant Learning of Continuous Spatiotemporal Dynamics, NeurIPS 2022.

---

> > > ### Author Response · Authors · 2023-08-21
> > >
> > > We genuinely thank you for carefully reading through our rebuttal and acknowledging the significance and novelty of our approach, especially compared to neural ODE/SDE methods.
> > > > Given that the horizon ℎ remains fixed once chosen, and considering that the training of the interpolator $\mathcal{I}$ is independent of the forecaster ${F}$, does this imply that alternative forecasting models capable of predicting $x_{t+h}$  from $x_t$ can also be employed to achieve continuous forecasting in conjunction with the trained interpolator $\mathcal{I}$?
> > >
> > > Yes, in principle, this would be possible.
> > >
> > > > If such is the case, it would be beneficial to conduct further ablation studies to elucidate the necessity and advantages of the forecaster  ${F}$.
> > >
> > > Please note the new figure 5 that we will include to our revised appendix (anonymous gdrive link: https://drive.google.com/file/d/1jFYdEn1tAkJ--HsndoXRndjJ2X_2qPTk/view?usp=sharing), where we ablate exactly this. Here, we show that the forecasts of $x_{t+h}$ of the forecaster $F$ gain more skill as the reverse diffusion process progresses (i.e. as the corresponding time of the diffusion step comes close from $t$ to $t+h$).
> > >
> > > >  For example, exploring diffusion settings in common practice could be insightful, i.e., investigating a forecaster that generates  $x_{t+h}$  by denosing from Gaussian noise conditioned on $x_t$ might be worth considering
> > >
> > > We thank the reviewer for this idea, and agree that this is an interesting method to consider. Honestly, we did not think of this method since it has not been proposed or used before. We believe that future work can explore if such a method will actually perform well. In this paper, we focus on our own proposed method, common multi-step forecasting approaches, and video diffusion models as baselines. Notably, while video diffusion is the most straight-forward approach for applying conventional diffusion models to multi-step forecasting, it had not been applied before to this problem.
> > >
> > > > (..) given the lack of evidence indicating that the intermediate $\hat{x_{t+i_n}}$ s are utilized
> > >
> > > This is wrong. When the optional line 6 of Alg. 2 is disabled, the intermediate $\hat{x_{t+i_n}}$ are used as forecasts for the intermediate timesteps between $t$ and $t+h$. This is what is actually being done in our SST dataset experiments. In addition, you can see in our ablations table (*No ref.* row in Table 5 in the appendix) that utilizing these intermediate $\hat{x_{t+i_n}}$ works well (but can sometimes be outperformed by enabling line 6 in Alg. 2).
> > >
> > > > As for neural SDE on high-dimensional data,I would like to suggest considering Neural-SPDE [5], which applies neural SDE to Navier-Stokes equations in a 64x64 grid setting. Its codebase provides implementations of Neural-SPDE as well as various baselines, including NCDE and NRDE. I recommend leveraging these baselines to showcase the effectiveness of DYffusion against neural ODE/SDE based approaches, if time permits for an empirical comparison.
> > >
> > > We thank you for the pointer to this baseline. In the limited time frame we had, we have done our best to run neural SPDE as a baseline, focusing on the spring mesh dataset since it has the lowest dimensionality amongst all our datasets. Please see the resulting figure in the following anonymous gdrive link: https://drive.google.com/file/d/1IoQuNvNKAphrLbBX7zUHTPsd2V5S96Po/view?usp=sharing
> > > This figure is the same as Fig 7b) of our joint rebuttal PDF, but extended by your proposed neural SPDE baseline. As you can see, our baselines and DYffusion outperform it.
> > > To attain this result, we run neural SPDE with ``n_iter=1, modes1=100, modes2=100, hidden_channels=32, solver=’fixed_point’ ``and a training horizon of ``h=100``. We note that increasing ``n_iter`` leads to significantly worse performance, and similarly for reducing the number of modes. We also would like to note that to our understanding neural SPDE models the noise W to be a sample path from a Wiener process, but does not really model the distribution of the forecast. Thus, the resulting forecasts are deterministic. Lastly, neural SPDE shines when the noise forcing is known (in their paper results $u_0 \rightarrow u$, where the noise $\xi$ is unknown, neural SPDE does not perform significantly better than baselines), which is not the case for our datasets or can be expected in real-world applications.

---

### Official Review · Reviewer_XpdM · 2023-07-04

**Soundness:** 2 fair
**Presentation:** 1 poor
**Contribution:** 3 good
**Rating:** 7
**Confidence:** 3

**Summary:**

In this paper, the authors propose to build on diffusion model for modelling spatio-temporal data.
In particular, they first train a time-dependent interpolation network which learn to interpolate the temporal dynamics given a frame at the horizon time $x_{t+h}$, a frame $x_t$, and an index $i$ interpolating between the two distributions.
Then, the idea is to frame a forecast model as a denoising model between $x_t$ and $x_{t+h}$, that is given an interpoled frame to be able to recover the original 'denoised' $x_{t+h}$ frame.
This approach allows for memory efficient multi-step predictions, as opposed to existing diffusion models for videos.
They apply the developed model to the forecasting of sea surface temperatures, Navier-Stokes flows, and spring mesh systems.

**Strengths:**

- The main advantage of the proposed approach seems to be the small memory footprint as it is constant w.r.t. the horizon, in contrast with standard dynamics forecasting (if I understand correctly).
- It also allows for 'upscaling' the resolution, or for making prediction at any time value, thus is not constrained to a regular discretisation of the time axis.
- The experiment on the sea surface temperatures dataset is quite promising, as the proposed method is able to accurately forecast not only the predicted mean but also the uncertainty with a lower computational cost than the video diffusion model MCVD. It would be valuable to explore whether one could obtain better performance at the expense of more computational cost.

**Weaknesses:**

- I found the writing to be confusing at times, I think that Section 3 could be better conveyed, in particular the choice of notations, but also the relation with diffusion models, and the 'cold posterior' sampling scheme.

**Questions:**

- Equation 3: How is the architecture handling the two inputs $x_t$ and $x_{t+h}$? By doubling the number of input channels?
- line 119: Why do we need this? This deserves more explanation.
- Equation 4: Why do we need the interpolator network here? Can't we directly predict $s^{(n+1)}$ with $F_\theta(s^{(n)}, i_{n+1})$? Aren't we actually extrapolating with the interpolator here?
- Algorithm 2: What is happening line 4? How does this iteratively refine the prediction of $x_{t+h}$? This is likely worthy of some more explaination.
- Equation: Isn't this ODE implying that the dynamics is Markovian?
- line 249: What is the Continuous Ranked Probability Score (CRPS)? Worth at least giving a high level idea I think.
- line 268-269: Why not training the DDPM with more diffusion steps? Since the number of steps is an hyperparameter for all methods it would be easier to compare them with the same number of steps, albeit could be varied to see how performance vs computation evolves.
- Table 1: Would you know why the performance of the models is widely different betweent the SST and the Navier-Stokes datasets?

---

> ### Author Rebuttal · Authors · 2023-08-10
>
> Thank you for the positive comments regarding the memory efficiency, continuous-time forecasts, and SST experiments of our work, as well as your valuable feedback to which we respond below.
>
> **Q1:** _Confusing section 3_
>
> **A1:**
> > I found the writing to be confusing at times, I think that Section 3 could be better conveyed, in particular the choice of notations, but also the relation with diffusion models, and the 'cold posterior' sampling scheme.
>
> Thank you for bringing this to our attention. We will add a glossary to the appendix that contains all our choice of notations in a single place. Could you let us know which specific parts were confusing? We would like to address these.
> Additionally, we will write out the objective for a conventional diffusion model (Eq. (1)) alongside our corresponding objective out (Eq. (3)) to make the connection in terms of analogous objectives more clear.
>
> Concretely, for a "generalized diffusion model" [2] (equation 1):
> $$||R_\theta(D(\mathbf{s}^{(0)}, n), \mathbf{x_t}, n) - \mathbf{s}^{(0)}||^2 = ||R_\theta(\mathbf{s}^{(n)}, \mathbf{x_t}, n) - \mathbf{s}^{(0)}||^2,$$
> where $\mathbf{s}^{(0)} = \mathbf{x_{t+1:t+h}}$, and $\mathbf{s}^{(n)}$ is a noisy version of $\mathbf{s}^{(0)}$ (the level of noise increases with $n$).
>
> And for DYffusion (equation 3):
> $$||F_\theta(\mathcal{I_\phi}(\mathbf{x_t}, \mathbf{x_{t+h}}, {i_n}), {i_n}) - \mathbf{x_{t+h}}||^2
> = ||F_\theta(\mathcal{I_\phi}(\mathbf{x_t}, \mathbf{s}^{(N)}, {i_n}), {i_n}) - \mathbf{s}^{(N)}||^2
> = ||F_\theta(\mathbf{s}^{(n)}, {i_n}) - \mathbf{s}^{(N)}||^2,$$
> where $\mathbf{s}^{(N)} = \mathbf{x_{t+h}}$ and $\mathbf{s}^{(n)} \approx \mathbf{x_{t+i_n}}$ is now a stepped backward in time version of $\mathbf{s}^{(N)}$. Note that the diffusion step indexing (superscript $n$) for DYffusion is reversed so that it aligns with the temporal indexing (subscript $t$), such that e.g. $\mathbf{s}^{(n)}  \approx \mathbf{x_{t+i_n}}$ temporally precedes $\mathbf{s}^{(n+1)}  \approx \mathbf{x_{t+i_{n+1}}}$. Accounting for the opposite order of indexing, the similarity between both approaches becomes clear.
>
> >  the 'cold posterior' sampling scheme
>
> The cold sampling algorithm is directly taken from [2] (their Alg.2; only the notation is adapted), and thus we refer the reader to [2] for intuition on how/why it works. For example, an alternative to sample from DYffusion (or any "generalized diffusion model") would be to replace line 4 in Alg. 2 with simply: $\mathbf{x_{t+i_{n+1}}} = \mathcal{I_\phi}(\mathbf{x_t}, \hat{\mathbf{x_{t+h}}}, i_{n+1})$. This corresponds to the _naive sampling_ algorithm of [2] (their Alg. 1) which performs worse than cold sampling as shown in [2] and in our new Table 10 in the joint rebuttal PDF. We will make this more clear in our revised version.
>
> **Q2:** _More questions_
>
> **A2:**
> > Equation 3: How is the architecture handling the two inputs $\mathbf{x_t}$ and $\mathbf{x_{t+h}}$? By doubling the number of input channels?
>
> Yes, exactly. We have updated the text to mention this explicitly.
>
> > line 119: Why do we need this? This deserves more explanation.
>
> Essentially, the look-ahead loss term simulates one step of the sampling process (Alg. 2) and backpropagates through it, so that the network is trained with an objective that is closer to (partially mimics) the sequential sampling process.
>
> > Equation 4: Why do we need the interpolator network here? Can't we directly predict  $\mathbf{s}^{(n+1)}$ with $F_\theta(\mathbf{s}^{(n)}, i_{n+1})$? Aren't we actually extrapolating with the interpolator here?
>
> $F_\theta$ always forecasts $\mathbf{x_{t+h}}=\mathbf{s}^{(N)}$ (see line 109), so we cannot use it to predict $\mathbf{s}^{(n+1)}$. As noted in line 132, $\mathbf{s}^{(n+1)}$ corresponds to $\mathbf{x_{t+i_{n+1}}}$. We will make clear that for all $n \in \{1, ..., N-1\}$ it holds that $0 < i_n < h$, so that $\mathbf{s}^{(n+1)}=\mathbf{x_{t+i_{n+1}}}$ is always between $\mathbf{x_{t}}$ and $\mathbf{x_{t+h}} \approx F_\theta(\mathbf{s}^{(n)}, i_{n+1})$, and thus the interpolator can be used to interpolate the timestep $t+i_{n+1}$.
>
> > Algorithm 2: What is happening line 4? How does this iteratively refine the prediction of x_t+ℎ? This is likely worthy of some more explaination
>
> See the last part of our response **A1** (regarding _"the 'cold posterior' sampling scheme"_). Essentially, it allows for more robust sampling over naive sampling defined above.
>
> > Equation: Isn't this ODE implying that the dynamics is Markovian?
>
> No. We assume you are referring to equation (5) or (6): The dynamics is not Markovian because each step in the prediction is dependent on $\mathbf{x_t}$ (initial state), in addition to the previous state $\mathbf{x_s}$.
>
> > line 249: What is the Continuous Ranked Probability Score (CRPS)? Worth at least giving a high level idea I think.
>
> The CRPS is a common metric for probabilistic forecasts [12, 19, 48, 50, 57]. It penalizes both absolute skill as well as sharpness of the ensemble of forecasts. It rewards small spread (i.e. sharpness) if the forecast is accurate, and measures the difference between the forecasted and observed cumulative distribution function (CDF). For a deterministic forecast, the CRPS reduces to the mean absolute error. We will add a more detailed explanation to our revised paper.
>
> > line 268-269: Why not training the DDPM with more diffusion steps?
>
> We tried 1000 diffusion steps for the DDPM as well, but found it decreased the performance  on the SST dataset.
>
> > Table 1: Would you know why the performance of the models is widely different between the SST and the Navier-Stokes datasets?
>
> We are also intrigued by this observation. Our current hypothesis is that the limited dataset size of the Navier Stokes/spring mesh dataset could be an important factor. Indeed, conventional diffusion models are data hungry since they need to learn how to map a noise distribution to the high-dimensional data distribution. We hope to investigate this more in the future.

---

> > ### Comment · Reviewer_XpdM · 2023-08-21
> > **response**
> >
> > Thanks for the response to my comments!
> >
> > > Could you let us know which specific parts were confusing? We would like to address these.
> >
> > I think it's partly due to the various indices $h$, $i_n$, $N$ etc. What's more it took me a 2nd read to eventually understand properly the interplay between the _forecaster_ and the _interpolator_. Thus I'd focus on this.
> >
> > > 'cold posterior' sampling scheme
> >
> > I had a closer read at the related paper and things are now clarified. The additional results from Table 10 also brings additional empirical evidence for using this scheme. I still believe that it's important to introduce this scheme a bit more than in the submitted manuscript.
> >
> > I've updated my score to reflect the clarifications and the fact that part of my concerns have been addressed. The main one remaining being clarity and presentation, yet this is tricky to address without being able to update the manuscript. I hope that the authors would improve on this so that most can benefit from this submission.

---

> > > ### Author Response · Authors · 2023-08-21
> > >
> > > We genuinely thank you for your feedback and taking the time to read through both our clarifications and the related paper introducing 'cold sampling'. We will introduce this scheme more in our revised paper, and use the easier to understand 'naive sampling' counterpart as a starting point for introducing the better performing cold sampling.
> > >
> > > > What's more it took me a 2nd read to eventually understand properly the interplay between the forecaster and the interpolator. Thus I'd focus on this.
> > >
> > > Thank you for your feedback! Do you think that Figure 6a) in our appendix makes the interplay more clear, potentially extended by explicitly adding the interpolator and forecaster symbols to their respective arrows? Would be beneficial to add it to the main text?

---

### Official Review · Reviewer_7msc · 2023-07-06

**Soundness:** 2 fair
**Presentation:** 3 good
**Contribution:** 2 fair
**Rating:** 5
**Confidence:** 4

**Summary:**

This paper proposes a new forecasting model for spatiotemporal data. The idea is based on separately training an interpolator and a forecaster network and applying them in an alternating fashion at inference time to iteratively refine the forward prediction. The inference procedure loosely resembles the denoising process in a diffusion model, where one computes a new denoising target and moves slightly towards it at every step. Numerical experiments are conducted on several datasets to validate its skills.

**Strengths:**

* The idea of iteratively refining forward predictions is interesting and quite novel to my knowledge.
* The paper is generally well written and easy to follow.
* The numerical experiments cover multiple non-trivial tasks and show moderate improvement in the metrics against benchmarked diffusion-based and ensemble prediction models.

**Weaknesses:**

* Although advertised as a diffusion model, the actual connection (at least to diffusion-based generative models) is handwavy at best. It is not clear that the underlying process arising from the defined noising and denoising operations form a well-defined generative model (for example see how DDPM is derived with clearly defined conditional and marginal distributions at each step). Only the variational posterior is provided in equation (4).
* The model is mostly deterministic (except for the last step that injects some Gaussian noise) so I think it’s important to show comparison against deterministic forecast models. The high memory footprint for multi-step training is somewhat fair, but there exist ways to improve single-step models (e.g. noise injection, curriculum training, etc.) which should be considered when validating the proposed model. This is necessary to justify doing the additional steps of interpolation.

Overall I feel in its current state given the strengths and weaknesses I vote for borderline reject tentatively.

**Questions:**

* Equation (4), the second line is not a probability. Do you mean a $\delta$ distribution at the interpolator output?
* Also equation (4), adding Gaussian noise only at the last step looks a bit cryptic to me. What is this trying to achieve besides adding randomness to the prediction and how is the noise level $\sigma$ determined?

**Limitations:**

The authors mention that input and output space must be the same and that inference costs are higher than predicting directly.

---

> ### Author Rebuttal · Authors · 2023-08-10
>
> We would like to thank you for the positive comments regarding the novelty, clarity, and experiments of our work, as well as your valuable feedback to which we respond below.
>
> _**Potential misunderstanding:**_ We would like to point out that there seems to be a key misunderstanding regarding our method being deterministic while it is actually probabilistic . As  noted in our problem statement section (see Setup, lines 65-67) we specifically study the problem of _probabilistic_ forecasting. Consequently,, our interpolator network,  $\mathcal{I_\phi}$, is designed to produce stochastic outputs (see the abstract, Algorithm 1 (Stage 2, line 1), and lines 106 & 111). We achieve this by enabling dropout at inference time. We show that this is a key component of our framework in the last two ablation rows (No. Dr, and No Dr. & $\sigma_\epsilon$) of Table 5 in the Appendix.
> Based on this misunderstanding, we will strive to make our generative formulation more clear in the revised methodology section as mentioned in answer A3 below.
>
>
> **Q1:** _Connection to conventional diffusion models_
>
> **A1:**
> > Although advertised as a diffusion model, the actual connection (at least to diffusion-based generative models) is handwavy at best. It is not clear that the underlying process arising from the defined noising and denoising operations form a well-defined generative model
>
> Our work builds upon cold diffusion [2], which "_paves the way for generalized diffusion models that invert arbitrary processes._" The cold sampling algorithm, which we use to sample from DYffusion (see Alg. 2), is a generalization of DDIM sampling for "generalized diffusion models" (see Appendix A.6 of [2]) and a key ingredient to make DYffusion work (see Table 10 of the joint rebuttal PDF). Our proposed forward and reverse processes are specifically designed to fall under this framework.
>
> *Given this context, it becomes clear that DYffusion is a generative model for forecasting that falls in the category of "generalized diffusion models."* We can see how this relationship may not have been clear in the text, and have added a new paragraph discussing these connections to our appendix.
>
> >  DDPM is derived with clearly defined conditional and marginal distributions at each step. Only the variational posterior is provided in equation (4).
>
> You are correct that  DYffusion loses some of the benefits of using a simple Gaussian distribution as the forward process. This is because DYffusion's forward process is based on a stochastic interpolator network, $\mathcal{I_\phi}$. We believe that this is a fair trade-off to make, especially because ultimately it is the posterior that we care about for forecasting. Future work can explore regaining some of the benefits from DDPM/DDIM.
>
>
> **Q2:** _Comparison against deterministic models_
>
> **A2:**
> > The model is mostly deterministic (except for the last step that injects some Gaussian noise) so I think it’s important to show comparison against deterministic forecast models
>
> As mentioned in the "potential misunderstanding" paragraph above, DYffusion is NOT a deterministic model. Its main source of stochasticity comes from the forward process/interpolator network. As such, deterministic models are not an adequate baseline and metrics such as CRPS and SSR are necessary for properly evaluating the probabilistic forecasts. In addition, we would like to highlight that our Perturbation baseline forecasts deterministically (see lines 649-653) and allows for the assessment of an ensembled deterministic model you might be looking for. **Also, please see the discussion of Fig. 7 in the joint rebuttal and PDF where we compare against a deterministic single-step model.**
>
> > The high memory footprint for multi-step training is somewhat fair, but there exist ways to improve single-step models (e.g. noise injection, curriculum training, etc.) which should be considered when validating the proposed model.
>
> Thanks for the suggestion. Noise injection and curriculum training have been only studied in the context of deterministic forecasting, not probabilistic forecasting as our paper. These techniques are orthogonal to the contributions proposed by this work to address multi-step forecasting challenges. For example, noise injection is a complementary method that can be added to further improve DYffusion too.
>
> While we would appreciate a reference for "curriculum training", we believe that you are thinking of approaches like the one used by [35]. It is one particular kind of performing multi-step training. In this paper, we choose the more common approach of directly predicting any of the multiple steps in a single forward pass (i.e. without needing to backpropagate gradients recursively as in [35]). An extensive benchmarking on how to best perform multi-step training is beyond the scope of this paper.
>
> **Q3:** _Equation (4)_
>
> **A3:**
> > Equation (4), the second line is not a probability. Do you mean a \delta distribution at the interpolator output?
>
> As mentioned in the "potential misunderstanding" paragraph above, the interpolator network is designed to be stochastic by enabling dropout at inference time. We will make this more clear in our writing.  For example, we have updated the text to make it explicit that $\mathcal{I_\phi}$ depends on a random variable $\xi$ (here, the randomly dropped out weights of the neural network) by writing $\mathcal{I_\phi}(\mathbf{x_t}, \mathbf{x_{t+h}}, i | \xi)$.
>
> > Also equation (4), adding Gaussian noise only at the last step looks a bit cryptic to me. What is this trying to achieve besides adding randomness to the prediction and how is the noise level  determined?
>
> We added Gaussian noise to make our Eq. (4) be consistent with Eq. (10) of DDIM [60]. However, in practice (see our Alg. 2) we do not add any Gaussian noise in the last step (as in DDIM). This is just of notational nature, so wee will remove this part in our revised version.

---

> > ### Comment · Reviewer_7msc · 2023-08-17
> >
> > Thanks for your clarifications and I apologize for not registering the obvious stochasticity introduced by inference dropout in the interpolator.  That said, I do not fully agree that deterministic models are irrelevant here as (a) comparison is done against MSE which is a deterministic metric and (b) probabilistic predictions may be obtained from deterministic models using ensemble with perturbed initial conditions. Uncertainties in probabilistic predictions inevitably contain both model errors and internal variability of the system and the latter is reflected in a setup like (b) to the very least.
> >
> > I have another follow up question regarding inference speed: how does the cost of the interpolator compare to that of the forecaster? The reason I am asking is that it is more fair to compare performance under the same constraint. For example, one might argue that the fair comparison would be to run an autoregressive model with half the step size (assuming that the autoregressive stepper is similar to the forecaster and interpolator alone). Superior metrics in this benchmark would be much stronger evidence that the guessing long into the future and iteratively refining it is indeed better than looking short-term only.

---

> > > ### Author Response · Authors · 2023-08-21
> > >
> > > Thanks for acknowledging the misunderstanding. We hope that you find our manuscript, and especially its originality and significance to the community, more convincing now.
> > >
> > > > That said, I do not fully agree that deterministic models are irrelevant here as (a) comparison is done against MSE which is a deterministic metric and (b) probabilistic predictions may be obtained from deterministic models using ensemble with perturbed initial conditions. Uncertainties in probabilistic predictions inevitably contain both model errors and internal variability of the system and the latter is reflected in a setup like (b) to the very least.
> > >
> > > We agree. For point (a), please see our new comparison against the deterministic models from [43] in Fig. 7 of our joint rebuttal PDF. For point (b), please note that we explicitly include this baseline in our Table 1, see the ``Perturbation`` row.
> > > > I have another follow up question regarding inference speed: how does the cost of the interpolator compare to that of the forecaster?
> > >
> > > This depends on the choice of the corresponding network architectures. Usually, the cost of the interpolator will be at most that of the forecaster. This is because interpolation is an easier task than forecasting, so we can expect to do fine by using an architecture with the same or lower complexity as the forecaster. In our experiments and for simplicity, all interpolator and forecaster networks share the same architecture for the respective datasets. As noted in the appendix B.5.1, we do halve the hidden dimensions of the interpolator relative to the forecaster network on the SST dataset.
> > >
> > > > The reason I am asking is that it is more fair to compare performance under the same constraint. For example, one might argue that the fair comparison would be to run an autoregressive model with half the step size (assuming that the autoregressive stepper is similar to the forecaster and interpolator alone). Superior metrics in this benchmark would be much stronger evidence that the guessing long into the future and iteratively refining it is indeed better than looking short-term only.
> > >
> > > We understand your concern. Unfortunately, we already train/evaluate on the highest possible temporal resolution, so it is not possible to half the step size. That being said, this proposed baseline would be effectively training on a horizon of half the length of before, and requiring twice the number of autoregressive steps for our full rollout evaluations on Navier-Stokes and spring mesh. The potential enhancement of rollout metrics through such an approach remains uncertain.
> > >
> > > It is worth noting that we have duly addressed and demonstrated that DYffusion operates at a slower pace compared to certain single-forward pass baselines, as substantiated by Table 1 and our discussion within the limitations section. Notably, the reassuring aspect lies in the fact that DYffusion's heightened computational demands during inference, though still more efficient than conventional diffusion models, consistently correlate with improved predictive performance in comparison to the established baselines.

---

> > > > ### Comment · Reviewer_7msc · 2023-08-21
> > > >
> > > > Thanks for your response. Despite still having my own reservation about the last point, I will raise my score to 5.

---

### Official Review · Reviewer_WiMo · 2023-07-06

**Soundness:** 2 fair
**Presentation:** 3 good
**Contribution:** 3 good
**Rating:** 7
**Confidence:** 3

**Summary:**

In this paper, authors tackle the long-term forecasting problem applied to dynamics system. To solve this problem, they propose to use diffusion principle along with interpolating and forecaster mechanisms. The former interpolates timestep data in between lookback and target windows (therefore, at a lower resolution).
Authors evaluate their proposal on three datasets, analyze and interpret the results as well as further discuss directions for improvements.

**Strengths:**

 * Interesting approach
 * Extensive ablation study, with summary in the main paper section
 * Interesting discussion of the results

**Weaknesses:**

 * Reproducibility (no code, Navier-Stokes and Spring-Mesh experiment set-ups are presented in the table caption, but I wonder if they are complete for anyone who would like to re-run the same experiments)
 * Comparison with the main reference [43] is in my opinion very subjective and should be improved (as authors build-up on [43] for two datasets, it seems to me that [43] should be included as a baseline)
 * Require some proof-reading


**Questions:**

The proposal is very interesting and sound promising. However, in my opinion, we are missing a representation of the proposed architecture to better understand the paper and how each module interacts with each other.

## Reproducibility
The details for training /validation and testing for Navier-Stokes (NSFlow) and Spring-Mesh (SM) are not provided in the dataset description, but in the experiments caption. And yet we can wonder if they are complete, which could make more difficult to reproduce results.

## Performance comparison
I think we are missing the time column in table 2 and the one for NSFlow experiment in Table 1.

`For the out-of-distribution test set of the Navier-Stokes benchmark, the results are almost identical to the one in Tab. 1, so we do not show them.` Why not include them in the appendix and let readers be the judge and make their own opinion?

## Comparison with reference [43]
As authors build up on ref. [43] models for NSFlow and SM, why are they not comparing with the model (unet and CNN) from such a reference. Especially looking at results from [43], for NSFlow from Figure 5, unet seems to achieve MSE between 0.05 and 0.007. These variations depend on the number of obstacles and there might be some difference in the set-up, but it would be good to see how DYffusion do compared to these baselines. As is it does not look like to me that the proposed model is doing better contrary to what authors are claiming. Other readers might feel the same and wonder what is the advantages of DYffusion. As a consequence, it would be good to define precisely the condition of NSFlow and SM and compare with original model.

`It is worth noting that our reported MSE scores are significantly better than the ones reported in [43]` I have difficulty agreeing with this claim. First, results in [43] are not presented in a table format so it is more difficult to compare. Authors should either provide their results with the same representation (stepMSE vs. OoD MSE) for instance in the appendix, for reader to better judge and compare both solutions. Or auhtors should include [43] models as mentioned above as a baseline in their table 2 (or both). While clearly setting the configuration of the forecast to make sure that we are dealing with similar set-up.

`our MSE results significantly surpass the ones reported in Fig. 8 of [43]`: Again, if authors wish to compare their results with model from [43], why not running forecast with CNN from reference [43] in the same condition? As Authors, in this paper, are re-using CNN from ref. [43], it should then be possible and prevent readers to go check paper [43] to make sure the settings are the same and try to grasp the similitude between papers as each paper use different representations of the results. In addition, if CNN performed so poorly in [43], why choosing it as the base model for SM dataset? Why not choose an MLP or nn kernel that seems to perform better on the figure mentioned by authors?

## Model evaluation
`resulting forecast to be visibly pleasing and temporally consistent` Not really a scientific judgement in my opinion…But it is indeed quite similar. Nevertheless, at t=3.70, the velocity of sample 2 to 5 are not really “visibly pleasing”, how should we interpret this?

## Ablation study
In my understanding, the proposal relies on the forecaster and interpolator. Therefore, I would have expected to see an ablation study, without these modules to judge of their importance. Is it impossible to do so?
Is it not possible also to do ablation study with different frame number of the interpolator (k-1)? BTW, unless I missed it, we don’t know what the value of k for each dataset is. Does it have an impact on the performance?

## Proof-read
I found some points to be corrected and authors should pay some more attention on possible other typos or issues:
 * Line 25. and and require -> and require
 * Line 92, should not it be x_{t+1}:{t+h}
 * Line 196, Sentence is a bit long, consider breaking it
 * Line 271, In the spring mesh dataset -> In the Spring Mesh dataset
 * Line 288, whether it is actually able -> whether it is actually possible
 * Line 302/303, For the Navier Stokes and spring mesh dataset it was sufficient -> For the Navier Stokes and Spring Mesh dataset, it was sufficient
 * Line 327, could be the reason for why -> could be the reason why


**Limitations:**

Authors discuss their results and limitations as well as future directions.

---

> ### Author Rebuttal · Authors · 2023-08-10
>
> We would first like to thank you for the positive comments and valuable feedback. We respond to your comments and questions below.
>
> **Q1:** _Reproducibility_
>
> **A1:**
> 1. **Code:** We have shared with the AC an anonymous link to our code for reproducibility. We will open-source our code and data when the manuscript is published (noted in the submission questions),
> 2. **Navier-Stokes and Spring Mesh set-ups:** We include instructions on how to reproduce our experimental results in the code README. We note that our evaluation procedure is *identical* to the benchmark dataset paper [43] (i.e. the same validation/test sets and the same metrics), except that we adapt it to probabilistic models (e.g. sample multiple forecasts per initial condition, compute CRPS and SSR). We will make it clear that we always train on the full training datasets of [43] and that we use the Navier-Stokes with 4 obstacles. We will note all these details in the revised appendix.
>
> **Q2:** _Performance comparison_
>
> **A2:**
> > Why not include Navier-Stokes out-of-distribution test set results in the appendix and let readers be the judge and make their own opinion?
>
> This is a good point. We have added these results to the revised appendix, and also show them in Table 10 of the joint rebuttal PDF.
>
> **Q3:** _Comparison with reference [43]_
>
> **A3:** We appreciate this feedback and have revised the manuscript to describe the experimental setup for Navier-Stokes and spring mesh in more detail (see response 2. of **A1** above).
>
> However, we would like to point out that [43] is a dataset paper. All baselines therein are ***deterministic*** models, and as such is not an adequate baseline for the ***probabilistic*** forecasting setup that we study in this paper.
> Nonetheless, **please see our Figure 7 in the joint rebuttal PDF, where we reproduce the relevant models from [43] and show that we substantially outperform them with all our baselines in terms of MSE**.
>
> Our Navier-Stokes and Spring Mesh ensemble mean MSE scores can be best compared to Fig. 10 (bottom; step prediction Unet with 4 obstacles) and Fig. 8 (bottom; step prediction CNN), respectively, of [43]. While the authors of [43] have not provided tabular results, comparing these figures reveals that 1) our reported MSE scores for Navier-Stokes ($0.022$) are competitive to the ones from Fig. 10 in [43], and 2) our spring mesh MSE ($4.74e-4$) is much lower than any reported MSE in Fig. 8 of [43], where none of the step prediction models achieve MSE less than $10^{-2}$. We have reached out to the authors from [43] to get exact values for these results so that we can report them in our paper as supplementary results.
> While helpful, this comparison is insufficient on its own, as MSE can only be part of the evaluation of a probabilistic forecasting model, since it does not capture the skill of the forecasted distribution like the metrics CRPS and SSR.
>
> Additionally, please note that our Perturbation baseline can be used by the reader to assess how direct applications of the models from [43] perform on our probabilistic skill evaluation. Indeed, except for the baselines being trained to forecast multiple steps, our Perturbation baseline, which forecasts deterministically (see lines 649-653), for Navier-Stokes is identical to the UNet used in [43].
>
> >  In addition, if CNN performed so poorly in [43], why choosing it as the base model for SM dataset? Why not choose an MLP or nn kernel that seems to perform better on the figure mentioned by authors?
>
> Diffusion models usually use UNet backbones, so to stay as close as possible to that, we chose the CNN model. In addition, our results show that the spring mesh CNN can actually perform very competitively when employing stochastic multi-step training as done for our baselines.
>
> **Q4:** _Model evaluation_
>
> **A4:**
> > _"resulting forecast to be visibly pleasing and temporally consistent"_. Not really a scientific judgement in my opinion
>
> We believe that a qualitative evaluation of the forecasted videos is important to complement the reported metrics.
>
> > Nevertheless, at t=3.70, the velocity of sample 2 to 5 are not really “visibly pleasing”, how should we interpret this?
>
> Such behavior is to be (sometimes) expected for long-rollouts of a probabilistic forecasting model. This is because we hope that the probabilistic model can capture any of the possible, uncertain futures (instead of forecasting their mean, as a deterministic model would do). As such it is reassuring that our samples show sufficient variation but also covers the ground truth (sample 1).
>
> **Q5:** _Ablation study_
>
> **A5:**
> > In my understanding, the proposal relies on the forecaster and interpolator. Therefore, I would have expected to see an ablation study, without these modules to judge of their importance
>
> The first sentence is correct. However, it is not possible to use our framework without both modules, as the sampling procedure relies on both. The Dropout baseline, i.e. a pure forecasting model, is the closest we can get to this.
>
> >  Is it not possible also to do ablation study with different frame number of the interpolator (k-1)? BTW, unless I missed it, we don’t know what the value of k for each dataset is.
>
> We have added a table ablating the choice of k for the SST dataset to the appendix and in Table 12 of our joint rebuttal PDF. We report the value of k for each dataset in lines 671-673, but we will also add it to a new row in the hyperparameters table 4.
>
> **Q6:** _Proof-read_
>
> **A6:**
> Thank you for helping pointing out several typos – we have fixed these. To clarify, We chose to write spring mesh in lowercase in the main text, because this is what the original dataset paper [43] does (while Navier-Stokes is capitalized).
> > Line 92, should not it be x_{t+1}:{t+h}
>
> No. We write $\mathbf{x_{t:{t+h}}}$ because we want to include the initial conditions $\mathbf{x_t}$ in this description (in addition to the to-be-forecasted snapshots $\mathbf{x_{t+1:{t+h}}}$).

---

> > ### Comment · Reviewer_WiMo · 2023-08-15
> >
> > Thank you for these clarifications.
> > I am satisfied with the answers, except with the comparison to [43]. Even if models in 43 are deterministic, if your goal is to have the best single- and multi-step predictions, your probabilistic model should beat probabilistic SOTA and deterministic SOTA. If you consider that deterministic and probabilistic are different domains, then don't mention that your model is doing better than 43 in the first place.
> > However, I would like to emphasis that all the details and additional explanations provided here should be included in the new revision of the paper (either in the main paper or appendixes).
> > Authors should give extra attention that all their equations, experiment set-ups and results be easily understandable for other to reproduce them if necessary.
> > Authors should also be careful of the typos (in the rebuttal document, in the caption of figure 7, the reference should be [43] not [44], right? Same for Table 12? Maybe you added a new reference which change the order and now 43 is 44, but you should double check it and adapt it in the Figure 7 legend).

---

> > > ### Author Response · Authors · 2023-08-18
> > >
> > > Thank you for your response and taking the time to carefully read our rebuttal. We are glad to hear that all answers except the following point satisfied you.
> > > > I am satisfied with the answers, except with the comparison to [43]
> > >
> > > It is unrealistic to expect a probabilistic model to ``beat’’ both probabilistic SOTA and deterministic SOTA because they are fundamentally different classes of models with very different objectives and evaluation criteria. A probabilistic SOTA is not optimized for point estimates, typically evaluated by MSE. Instead, probabilistic models emphasize on calibration and coveraged, as reflected in CRPS.  Therefore, they are not comparable to a deterministic SOTA. **However, we now see the benefit of including a direct comparison to the models from [43] in terms of MSE, which is why we will include the new figure 7 of the joint rebuttal PDF, plus discussion, into our revised paper**. *Please let us know if there is any other benchmarking against deterministic models or [43] that you think is important to include.*
> > >
> > > > if your goal is to have the best single- and multi-step predictions
> > >
> > > We would like to stress that our goal is NOT to "have the best single-step predictions", but rather we focus on multi-step probabilistic forecasting, and especially long rollouts in this paper. This is an important distinction, because while the models from [43] are appropriate for single-step predictive skill comparison, these models lack stability for long-term forecasts (the focus of this work). We demonstrate this in figures 7a and 7b in our joint rebuttal PDF, where the models from [43] diverge or significantly underperform compared with our method AND baselines in terms of MSE. This challenge is why we focus on multi-step/long-term probabilistic forecasting.
> > >
> > > > However, I would like to emphasis that all the details and additional explanations provided here should be included in the new revision of the paper (either in the main paper or appendixes).
> > >
> > > Rest assured that we will include all these details to our revised paper or appendix.
> > >
> > > > Authors should also be careful of the typos (in the rebuttal document, in the caption of figure 7, the reference should be [43] not [44], right? Same for Table 12? Maybe you added a new reference which change the order and now 43 is 44, but you should double check it and adapt it in the Figure 7 legend).
> > >
> > > Thank you for pointing this out, you are correct that [44]  in our joint rebuttal PDF is [43] from our submitted main paper. We apologize for this inconsistency that arises because we strived to integrate the reviewers feedback directly into our revised paper. In the final version this will not be a problem.

---

> > > > ### Comment · Reviewer_WiMo · 2023-08-18
> > > >
> > > > Thank you for these additional responses.
> > > >
> > > > > It is unrealistic to expect a probabilistic model to ``beat’’ both probabilistic SOTA and deterministic SOTA
> > > >
> > > > Perhaps, but that very important to have the global picture of where your proposal is beating SOTA (no matter the SOTA deterministic or probabilistic) and where it fails. Where it fails might be room for improvements.
> > > > Especially, in your rebuttal figure 7, even though your proposal is not beating ref. 43 on short-term horizon, it seems very close to it. Therefore, your proposal might be a good trade-off when dealing with both short- and long-term forecast. These types of results and comparison emphasis what you mentioned:
> > > > > we focus on multi-step probabilistic forecasting, and especially long rollouts in this paper. This is an important distinction, because while the models from [43] are appropriate for single-step predictive skill comparison, these models lack stability for long-term forecasts
> > > >
> > > > For anyone in the community not clear with difference between deterministic and probabilistic models, this kind of information is gold and on top of that it is back up by results. And Figure 7 makes this very clear.
> > > >
> > > > In my opinion, with such details you provide the full picture by doing a fair comparison and a clear conclusion: as you expected, your proposal excel in long-term horizon, while being competitive on short-term horizon.
> > > >
> > > > Considering that Authors will attach importance to include all these details to their manuscript, I increased my score from 5 to 7.

---

> > > > > ### Author Response · Authors · 2023-08-21
> > > > >
> > > > > We deeply appreciate your feedback and careful consideration of our rebuttal, resulting in a higher score. We believe that your comments have significantly improved our paper and allowed us to more effectively convey its significance to the community. As mentioned before, this will all be reflected in our final version of our paper.

---

> > > > > > ### Comment · Reviewer_WiMo · 2023-08-22
> > > > > >
> > > > > > I'm glad my comments could help Authors refine their submission.

---

### Author Rebuttal · Authors · 2023-08-10

We thank the reviewers for their thoughtful comments.

We are particularly encouraged by the reviewers finding our work  _”interesting”_ (WiMo and 7msc), _”quite novel”_(7msc) and _”quite promising”_ (XpdM).
We are glad to hear that reviewers found our paper _“well written and easy to follow”_ (7msc) with _“numerical experiments cover multiple non-trivial tasks”_ (7msc) and an _“extensive ablation study”_ (WiMo).

One common issue with our work was insufficient clarity of our methodology section (XpdM) which may have caused misunderstandings with regard to our method being “deterministic” (7msc) or requiring being benchmarked against deterministic baselines (WiMo).
As reviewer 4TEk correctly notes our work _“reimagines the continuous-time probabilistic forecasting problem as a diffusion process”_, and we would like to re-emphasize that our **method was specifically designed to _generate probabilistic_ forecasts**, which is why our baselines and evaluation procedure are for the probabilistic forecasting setting too.

In our joint rebuttal PDF we have added the following new figures and tables:
- Figure 7 shows that deterministic single-step forecasting as in [43] is substantially outperformed by all our own probabilistic multi-step baselines as well as DYffusion. In addition, you can observe that DYffusion performs especially well on long-range forecasts relative to the baselines. These results were requested by reviewer **WiMo** and should also be interesting to reviewer **7msc** who requested to “show comparison against deterministic forecast models”. To attain Fig. 7, we have reproduced the UNet and CNN baselines from [43]. The only difference between them and our Dropout-multi-step baseline (called “Dropout” in our paper) is that the latter is trained to forecast multiple steps and has inference dropout enabled.
- Table 10 shows that cold sampling (Alg. 2) substantially outperforms naive sampling, especially on the SST dataset.  Cold sampling was proposed by [2], while naive sampling is their generalization of DDPM sampling for “generalized diffusion models”. Naive sampling corresponds to replacing line 4 in Alg. 2 with: $\mathbf{x_{t+i_{n+1}}} = \mathcal{I_\phi}(\mathbf{x_t}, \hat{\mathbf{x_{t+h}}}, i_{n+1})$. This finding is consistent with [2] who find that naive sampling “works well for noise-based diffusion” but “yields poor results” for generalized diffusion models, and should be especially interesting to reviewers **7msc** and **XpdM**.
- Table 11 provides the ablation study requested by reviewer **WiMo** on ablating the value of k, i.e. the number of artificial diffusion steps used by DYffusion, for the SST dataset.
- Table 12 provides the Navier-Stokes out of distribution results requested by reviewer **WiMo**.

We are of the opinion that the valuable feedback provided by the reviewers has significantly enhanced the quality and clarity of our paper. As a result, the core contributions of our work, centered around the introduction of a pioneering framework for probabilistic dynamics forecasting and complemented by robust empirical evidence and an exhaustive ablative study, have been elevated.

---

### Comment · Area_Chair_vTU9 · 2023-08-21
**Final summary on the originality and significance of your work**

Dear authors,

Thank you for submitting your work to NeurIPS, responding to the initial reviews provided by the reviewers, and engaging in subsequent discussions with them during this author-reviewer discussion phase.

As you can see, even though we are approaching the end of this discussion phase after much deliberation, there are still divided opinions among the reviewers regarding the acceptance decision. As such, may I invite you to give a final summary in light of the further discussions with the reviewers? In particular, please address specifically the originality and significance of this work to justify its acceptance for presentation to the NeurIPS community. Originality and significance, among others, are major review criteria for acceptance, as you can see from the NeurIPS reviewer guidelines: https://nips.cc/Conferences/2023/ReviewerGuidelines.

We look forward to your summary (or “pitch”) soon.

Best,
AC

---

> ### Author Response · Authors · 2023-08-21
> **Pitch**
>
> ## Originality:
> Our proposed method, DYffusion, is a completely new approach for performing probabilistic multi-step spatiotemporal forecasting. This stands in a refreshing contrast to the most common approaches for spatiotemporal forecasting, which tend to focus on deterministic models and single-step training. These approaches have limitations:
> - Deterministic models are ineffective for uncertainty quantification. Perturbing the initial conditions can generate multiple forecasts from a deterministic model, but this underperforms in our results (see Perturbation baseline). Further, they are commonly trained on MSE loss, which leads to mode collapses, especially in long rollouts, and physically unrealistic or blurry forecasts.
> - Single-step forecasting training leads to a deviation between training and inference, since dynamics forecasting models are usually used for long rollouts and long-term forecasting. As a result, single-step forecasting models tend to collapse or underperform when used for long rollouts (see our Fig. 7 in the global rebuttal PDF).
>
> DYffusion presents a natural solution for both these issues, by designing a temporal diffusion model (leads to naturally training to forecast multiple steps) and embedding it into the “generalized diffusion model” framework so that by taking inspiration from existing diffusion models we can build a strong probabilistic forecasting model.
>
> In addition, DYffusion is the **first** diffusion model that relies on a task-informed forward/reverse process (here, we propose to use temporal interpolation and forecasting). All other existing diffusion models, albeit more general, use data corruption processes. As a result, our work gives a new perspective on designing a capable diffusion model, and may lead to a whole family of task-informed diffusion models.
>
> ## Significance:
>
> Accurate high-dimensional forecasts with calibrated uncertainty are of high practical relevance but challenging. DYffusion is the first diffusion model to employ dynamics-informed forward and reverse processes, effectively addressing both probabilistic and multi-step forecasting—often neglected by existing deep learning approaches. This pioneering solution yields strong empirical results and opens exciting avenues for further improvements.
>
> Our method shows significant improvement over the state-of-the-art video diffusion model in speed, memory footprint, and forecasting performance. DYffusion is the best or second-best model on all three benchmark datasets. It lowers the barrier for training/using diffusion models for dynamics forecasting, and performs strongly on challenging datasets where video diffusion models underperform or fail (here, Navier Stokes and spring mesh).
>
> ## Central concerns by the reviewers
>
> ### Misunderstanding by Reviewer 7msc
>  The main criticism from Reviewer 7msc is our model being  `` mostly deterministic’’ whereas our model is designed for **probabilistic forecasting**. They have acknowledged this misunderstanding in the comment. We also provided detailed answers to their follow-up questions on (1) comparison with deterministic models (2) inference speed.
>
> ### Baselines by Reviewer 4TEk
>  The central concern raised by Reviewer 4TEk is the lack of "baselines in neural ODE and neural SDE’’ because of DYffusion being "closely related to neural ODE and SDE”.
> The reviewer has acknowledged that significant differences between these approaches exist. To accommodate their request, we have run their proposed baseline, Neural SPDE, and shown that it underperforms both our method AND our baselines in terms of deterministic MSE. We would like to emphasize that this is to be expected because Neural SPDE assumes the noise underlying the data generating process to be a known and observed Wiener process. In our datasets and for real-world data this is a limiting assumption.
> It also assumes the noise be one sample path from the stochastic process, and hence models a deterministic function instead of a distribution. Thus, it is another inappropriate baseline for the probabilistic evaluation that we conduct in our paper.

---

> > ### Comment · Area_Chair_vTU9 · 2023-08-22
> >
> > Thanks for responding to my request and addressing the concerns from the reviews which includes conducting more experiments.
> >
> > AC

---

### Decision · Program_Chairs · 2023-09-21

**Decision:**

Accept (poster)

**Comment:**

This paper proposes a diffusion model for spatiotemporal forecasting, which goes beyond generating static images to dynamics forecasting by leveraging temporal dynamics learned from the data.

The proposed diffusion model is interesting. It can effectively reduce the memory footprint and achieve speedup as demonstrated by the experimental results. Unlike deterministic forecasting methods which only predict the mean, prediction uncertainty can also be quantified by the proposed model. The experiments reported in the paper are generally strong. Nevertheless, some concerns were raised by the reviewers in their initial reviews, which include both the technical content and the paper presentation. We thank the authors for responding to them and conducting further experiments to provide more details. Consequently, three of the four reviewers decided to increase their overall ratings after some of their concerns were clarified.

While we are inclined to recommending this paper for acceptance, we strongly recommend the authors to take all our comments and suggestions into consideration when revising the paper for a broader audience.